Manuscript for *Earth System Science Data*

# A Database of Aircraft Measurements of Carbon Monoxide (CO) with High Temporal and Spatial Resolution during 2011 – 2021

Chaoyang Xue[1], Gisèle Krysztofiak[1], Vanessa Brocchi[1], Stéphane Chevrier[1], Michel Chartier[1], Patrick Jacquet[1], Claude Robert[1], Valéry Catoire[1]*

[1]Laboratoire de Physique et Chimie de l'Environnement et de l'Espace (LPC2E), CNRS – Université Orléans – CNES (UMR 7328), 45071 Orléans cedex 2, France

Correspondence: Valéry Catoire (valery.catoire@cnrs-orleans.fr)

**Abstract**

To understand tropospheric air pollution at regional and global scales, the SPIRIT airborne instrument (SPectromètre Infra-Rouge In situ Toute altitude) was developed and used on aircraft to measure volume mixing ratios of carbon monoxide (CO), an important indicator of air pollution, during the last decade. SPIRIT could provide high-quality CO measurements with $1\sigma$ precision of 0.3 ppbv at a time resolution of 1.6 s thanks to the coupling of a quantum cascade laser to a Robert optical multi-pass cell. It can be operated on different aircraft such as Falcon-20 and ATR-42 from DLR (Germany) and SAFIRE (CNRS-CNES-Météo France). With support from various projects, more than 200 flight hours measurements were conducted over three continents (Europe, Asia, and Africa), including two inter-continental transects (Europe-Asia and Europe-Africa). Levels of CO and its spatial distribution are briefly discussed and compared between different regions/continents. CO generally decreases with altitude except in some cases, indicating the important contribution of long-distance transport to CO levels. A 3D trajectory mapped by CO level was plotted for each flight and is presented in this study (including a supplementary information). The database is archived on AERIS database (https://doi.org/10.25326/440), the French national center for atmospheric observations (Catoire et al., 2023). Besides, it could help to validate model performances and satellite measurements. For instance, the database covers measurements at high-latitude regions (i.e., Kiruna, Sweden, 68˚N) where satellite measurements are still challengeable and at low-latitude regions (West Africa and South-East Asia) where in situ data are scarce and satellites need more validation by airborne measurements.

# 1 Introduction

The study of atmospheric composition has been widely conducted and understood through comprehensive field campaigns and ground level air quality monitoring networks (including mountain summit- and tower- based platforms) in the world (e.g., Acharja et al., 2020; Andreae et al., 2015; Brown et al., 2013; Daellenbach et al., 2020; David et al., 2021; Guo et al., 2014; Hanke et al., 2003; Harrison et al., 2012; Ravi Kant Pathak et al., 2009; Ryerson et al., 2013; Sellers et al., 1995; Shi et al., 2019; Tang et al., 2021). The very recent fifth WHO Air Quality Database (https://www.who.int/data/gho/data/themes/air-pollution/who-air-quality-database, last access: July 18, 2023) compiles ground measurements on air quality for over 6000 cities/human settlements in more than 100 countries. This provides a large body of datasets to understand air quality, its health impacts, and atmospheric chemistry/dynamics in the lower atmosphere as well as its interaction with the biosphere, and to assess model performance in predicting near-ground atmospheric composition. However, those measurements are typically limited to the boundary layer, arising challenges and limitations in understanding the atmospheric chemistry and dynamics above this layer (i.e., free troposphere, stratosphere) and showing the necessity of airborne measurements. Airborne measurements are also necessary for the validation of satellite observations which have undergone fast development and shown important prospects in the past decades. However, measurements at high-latitude regions are still challengeable for satellite, and hence, validation of satellite in those regions are of significant necessity (Hegarty et al., 2022; Wizenberg et al., 2021). Moreover, many important gases for atmospheric chemistry, air quality, and global climate have too low abundance to be detected by satellite. The vertical profile and regional distribution of those species must be detected by in-situ airborne measurements.

Aircraft serving as a research platforms start in the late 1920's, beginning with meteorological measurements such as temperature and altitude in the UK (Gratton, 2012 and therein). Since the 1940's, scientists have been using aircraft to sample air to better understand the composition and related processes of the atmosphere far above the surface (https://earthobservatory.nasa.gov/blogs/fromthefield/2016/07/26/long-history-of-using-aircraft-to-understand-the-atmosphere/, last access: July 18, 2023). Owning to the development of rapid-response and high-precision instruments, aircraft measurements became more and more popular, particularly in North America and Europe (Ryerson et al., 2013; Fast et al., 2007; Hamburger et al., 2011; Fehsenfeld et al., 2006; Mallet et al., 2016; Andrés Hernández et al., 2021; Machado et al., 2018; Crawford et al., 2021). In addition to measurements within the boundary layer, aircraft platforms could allow measurements through the troposphere and even the lower stratosphere (< 20 km above sea level (asl)), through which understanding the atmospheric dynamics and the distribution of pollution at both horizontal and vertical scales could be achieved.

Carbon monoxide (CO) is an important atmospheric carbon compound, mainly emitted by incomplete combustion processes, e.g., biomass burning, fossil fuel combustion, etc. Its moderate lifetime of ca. 1-2 months indicates that its abundance is generally impacted by emissions at a regional scale. In polluted atmospheres such as urban areas, CO is mainly produced from

incomplete combustion processes such as coal/gasoline/diesel/biomass combustion, and its concentration could reach levels
of several ppmv (Xue et al., 2020; Dekker et al., 2019). In the background atmosphere (e.g., remote areas, marine boundary layer) or the upper troposphere, CO is generally low (< 100 ppbv) (Lelieveld et al., 2008) mainly resulting from atmospheric oxidization processes e.g., the oxidation of methane by OH radicals. A high level of CO in a background atmosphere typically indicates impacts of regional transport (or convection/eruption for the upper troposphere) (Krysztofiak et al., 2018). Therefore, aircraft CO measurements can provide an overview of the pollution in the studied region and are a good indicator of regional
pollution levels and intensive emission events, e.g., wildfires (Jaffe and Wigder, 2012; Jaffe et al., 2022). There are already available aircraft CO measurements, which provide a broader spatial distribution. Those are mainly from US projects, such as TRACE-P in 2001 (Palmer et al., 2003), INTEX-B in 2006 (Luo et al., 2007), GoAmazon2014/5 in Brazil (Machado et al., 2018), WE-CAN in 2018 (Permar et al., 2021), and FIREX-AQ in 2019 (Bourgeois et al., 2022), and European projects, such as DABEX in 2006 (Johnson et al., 2008), EUCAARI-LONGREX  and  APPRAISE-ADIENT in 2008 (McMeeking et al.,
2010), CLARIFY in 2017 (Haywood et al., 2021), EMeRGe-EU in 2017 and EMeRGe-Asia in 2018 (Forster et al., 2023), BLUESKY in 2020 (Voigt et al., 2022).

Another important utilization of aircraft measurements is the validation of chemical-transport model simulations. Satellite observations are more and more popularly used to study atmospheric composition. Many satellites can detect CO, such as MOPITT        (https://terra.nasa.gov/about/terra-instruments/mopitt,        last        access:        July        18,        2023),        IASI
(https://www.eumetsat.int/iasi, last access: July 18, 2023) and TROPOMI (http://www.tropomi.eu/, last access: July 18, 2023). However, satellite observations of CO need to be validated by measurement at different regions. For instance, recently, Wizenberg et al. (2021) compared CO measurements from TROPOMI, the Atmospheric Chemistry Experiment (ACE) Fourier transform spectrometer (FTS), and a high-Arctic ground-based FTS. They found different biases in different regions, e.g., positive biases (ca. +7%) in northern and southern polar regions and negative biases (ca. -9%) in equatorial regions.

Herein, based on a sensitive airborne instrument (SPIRIT) with high-resolution and high-precision CO detection, seven aircraft campaigns funded by European and French national projects were conducted worldwide between 2011 and 2021, accompanied by two inter-continental measurements. This paper summarizes all the data obtained during those aircraft campaigns.

## 2 Method: the SPIRIT Instrument

### 2.1 Set-up

In 2011, an infrared laser absorption spectrometer called SPIRIT (SPectromètre Infra-Rouge In situ Toute altitude) was developed at LPC2E for airborne measurements of trace gases in the troposphere. Details about this instrument can be found in Catoire et al (2017). Briefly, the coupling of a single Robert multi-pass optical cell (with a path length to be adjusted up to 167.78 m) with three interchangeable quantum cascade lasers (QCLs) was designed, which allows selecting trace gases to measure, according to the scientific objectives. Absorptions of the laser radiations by the species in the cell at reduced pressure

(<40 h Pa) are quantified using a HgCdTe photodetector cooled by the Stirling cycle, according to the Beer-lambert law. For CO, two absorption rovibrational lines were successively used, namely 2179.772 and 2183.224 cm$^{-1}$, with the functioning conditions for the lasers indicated in Table 1. CO was measured on all the scientific flights, regarding its essential role in atmospheric chemistry, as discussed in Section 1. Hence, in this database, all the airborne CO measurements during the last decade are summarized.


**Table 1: Parameters/performance of the SPIRIT instrument.**

| Years of Meas. | Spectral domain swept (cm$^{-1}$) | Current + ramp (mA) | T_QCL (ºC) | Precision (1σ, ppbv) |
|---|---|---|---|---|
| 2011-2016 | 2179.70-2179.85 | 600 + 13 | -12.5 | 0.3 |
| 2019-2021 | 2183.15-2183.35 | 595 + 15 | -13.2 | 0.3 |

**2.2 QA/QC (Quality Assurance / Quality Control)**

In Catoire et al (2017), laboratory experiments and in-flight intercomparisons with other instruments were conducted to assess
the performances of SPIRIT and the quality of the data. Besides, for each field campaign, in-flight calibrations were also conducted during every flight. It was noted that until 2016 (see Section 3.5), SPIRIT could provide high-quality CO data with a precision of 0.3 ppbv (1σ) and an overall uncertainty of less than 1 ppbv, when regular in-flight calibration vs. WMO standard were performed. This is still the case since 2016. As an example, in Section 3.7, we present data with in-flight calibrations.

**3 Results and Discussion**

**During the 2011-2021 period, SPIRIT was used in 7 aircraft campaigns with 74 scientific flights. In total, 208.8 hours of in-flight CO measurements were obtained (**

Table 2). Figure 1 summarizes the locations of all the measurements presented in this paper, i.e. over three continents, Europe, Asia, and Africa. Except for the SHIVA winter campaign, all campaigns were conducted in late spring or summer. In general, most campaigns were conducted in Europe, except the SHIVA and DACCIWA campaigns conducted in Southeast Asia and
West Africa, respectively. A wide area was covered by the measurements, for example, including tropical regions (SHIVA and DACCIWA) and northern mid-latitude regions (all others). Measurements during the MAGIC 2021 also covered a part of the northern high-latitude region. In general, the CO level depends on the pollution, with the highest values in the boundary layers (< 2 km altitude) of areas influenced by anthropogenic activities. The upper troposphere (> 6 km altitude) is cleaner (except for MAGIC 2021 high latitude area influenced by long range transport: see more detail in Sections 3.8 and 3.10). Also,

it is worth noting that two inter-continental flights, one from Europe to South Asia and the other from Europe to West Africa, were conducted during the transition flights for SHIVA and DACCIWA projects (more details in Section 3.9).

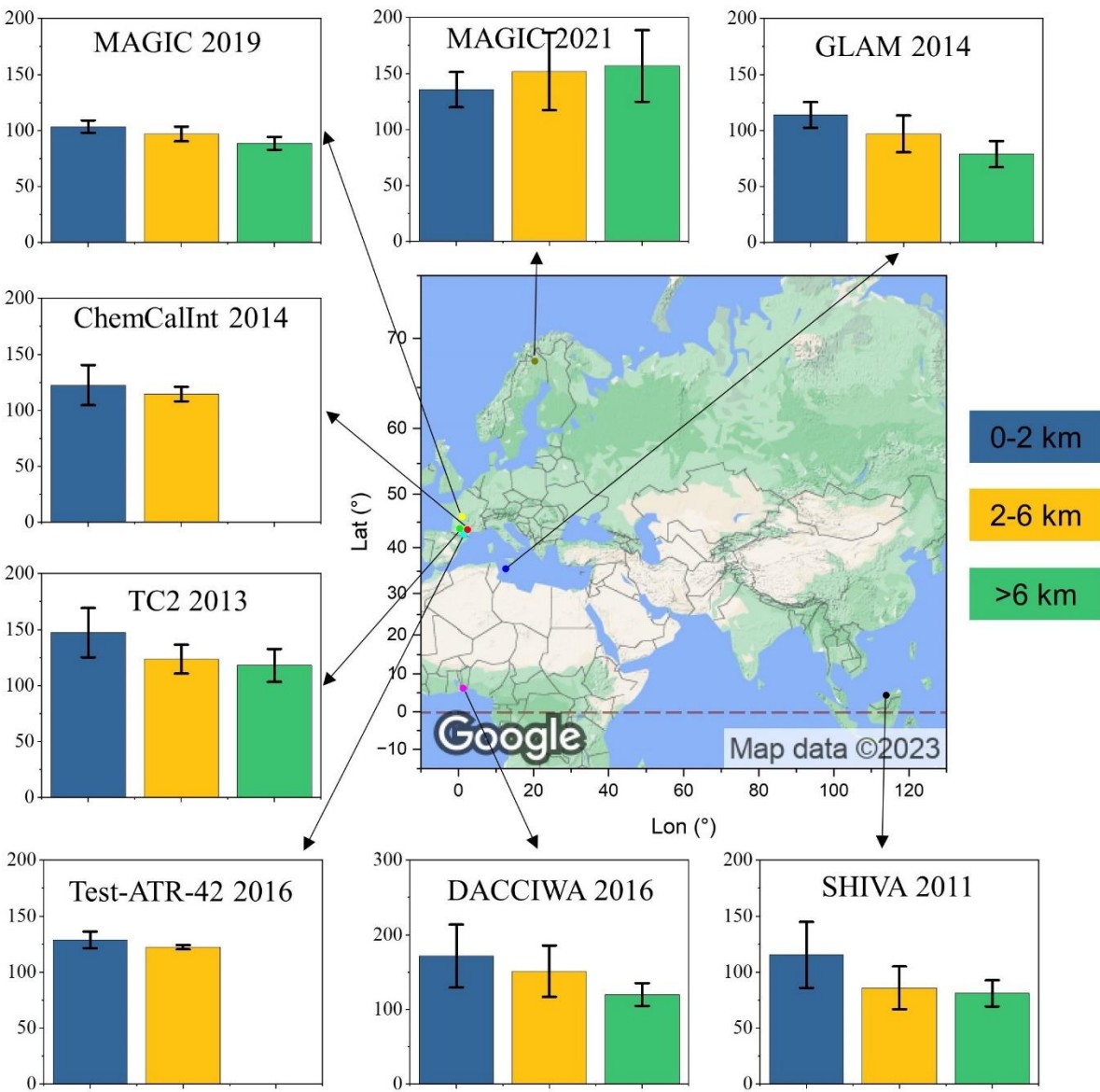

**Figure 1. Locations of the main airport used for each project. Map copyright: © GoogleMap. The same airport in Toulouse was used for TC2-2013, ChemCallnt-2014, and Test ATR-2016, then the overlapped points are offset and plotted horizontally. Lampedusa E**
**Linosa airport is selected for GLAM-2014 as it is in the center of the measurement area. Bar plots show the average CO levels in the boundary layer (0-2 km), lower free troposphere (2-6 km) and upper free troposphere (>6 km) for each project.**

**Table 2: Information about SPIRIT CO measurements.**

| Projects | Period | Aircraft | Number of Flights | Region | Duration (h) |
|---|---|---|---|---|---|
| SHIVA (2011) | Nov. – Dec. 2011 | A | 16 | 1º – 8º N, 114º – 122º E | 58.5 |
| SHIVA (2011)[a] | Nov. 2011 | A | 5 | 4º – 49º N, 11º – 115º E | 17.0 |
| TC2 (2013) | Mar. – May 2013 | B | 11 | 40º – 48º N, -8º – 9º E | 26.8 |
| ChemCalInt (2014) | May 2014 | C | 4 | 42º – 46º N, 0º – 6º E | 13.9 |
| GLAM (2014) | Aug. 2014 | B | 9 | 33º – 46º N, -1º – 35º E | 24.3 |
| Test sur ATR-42 (2016) | May 2016 | C | 1 | 42º – 44º N, 0º – 2º E | 2.3 |
| DACCIWA (2016) | Jun. – Jul. 2016 | A | 14 | 3º – 11º N, -5º – 3º E | 51.7 |
| DACCIWA (2016)[b] | Jun. 2016 | A | 5 | 5º – 52º N, -18º – 13º E | 13.3 |
| MAGIC (2019) | Jun. 2019 | B | 3 | 43º – 49º N, -2º – 4º E | 9.7 |
| MAGIC (2021) | Aug. 2021 | C | 6 | 64º – 69º N, 6º – 27º E | 21.4 |
| Total | | | **74** | | **208.8** |

[a]: 5 inter-continent flights from Europe to Asia; [b]: 5 inter-continental flights from Europe to Africa.

A: Falcon-20 DLR; B: Falcon-20 SAFIRE; C: ATR-42 SAFIRE

## 3.1 SHIVA – Malaysia (2011)

In the framework of the SHIVA (**S**tratospheric Ozone: **H**alogen **I**mpacts in a **V**arying **A**tmosphere) project of the European Commission FP7-Enviroment Program (https://cordis.europa.eu/project/id/226224, last access: July 18,2023), 16 research flights were conducted using the German Aerospace Agency (DLR) Falcon-20 aircraft in Malaysia (Borneo Island) between November 16 and December 11, 2011. The three-dimensional trajectories with CO volume mixing ratios (vmr) are displayed in the Supplementary Information in Figures S1 to S16. Measurements during the 5 inter-continental flights from Europe to Asia are presented in Section 3.9.

The SPIRIT instrument was on board during all the flights (58.5 hours) for measurements of CO vmr. Figure 2 shows all the trajectories of those flights. Most measurements were conducted along the coastal lines of Malaysia and Brunei, with several flights reaching the hinterland of this island and several flights crossing the South China Sea and reaching the region of Malacca Strait. The measurement domain represents a typical tropical region (latitude scale: 1º -7º N).

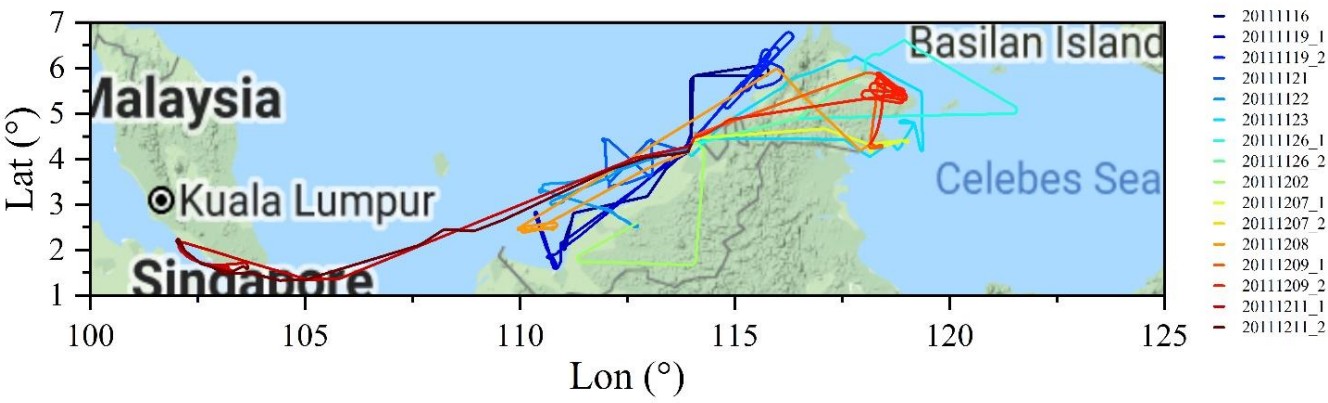

**Figure 2: Flight trajectories during the SHIVA 2011 campaign. Map copyright: © GoogleMap.**

Figure 3 exhibits the trajectory of the first scientific flight on November 19, 2011, which gives an insight of the CO level in the boundary layer and free troposphere of the Borneo island. The aircraft took off from Miri airport (Malaysia), gradually increasing its altitude to 8400 m asl. It flew along the coastal line of the Sarawak state (Malaysia) and reduced its flying height before reaching the southwest of the Sarawak state. Then, after conducting horizontal measurements over the ocean boundary layer, the aircraft increased its flying height and flew back to Miri airport. After about 2.4 hours of measurements, the aircraft landed at Miri airport. During the flight, the observed CO reached 140 ppbv in the boundary layer (0-2 km) and < 100 ppbv in the lower free troposphere (2-6 km), but relatively stable around 100 ppbv at ~8 km. Moreover, based on the measurements on flights referenced as 20111119_2, 20111209, 20111211_1, and 20111211_2 in the SHIVA project, Krysztofiak et al. (2018) found correlated enhancements of CO, $CH_4$, and short-lived halogen species (i.e., $CH_3I$ and $CHBr_3$) at heights around 11-13 km asl, which is interpreted as the fingerprint of the vertical transport from the boundary layer driven by the convective updraft. The fraction of air mass in the from the boundary layer can reach 67%, indicating the significant impact of the convective system on the composition of the upper troposphere in the tropical regions (Hamer et al., 2021).

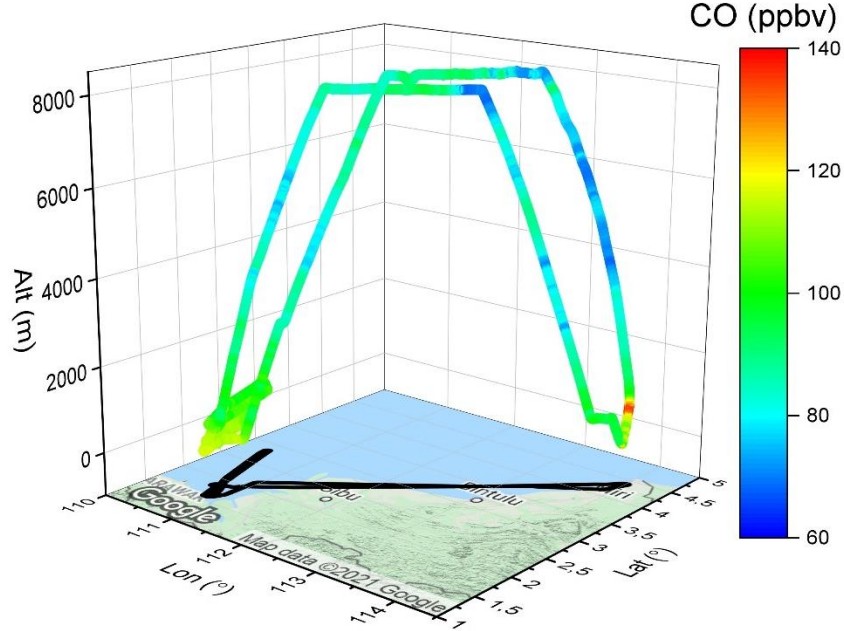

**Figure 3: Flight trajectory colored by CO levels on flight-SHIVA_20111119_1. Map copyright: © GoogleMap.**

### 3.2 Traînées de Condensation et Climat: TC2 - France (2013)

In the framework of the contrails and climate project TC2 (Each time the sampling inlet is connected to the cylinder containing the WMO CO standard de Condensation et Climat, http://www.cerfacs.fr/TC2/index.html, last access: July 18,2023), 11 flight measurements (Figures S17-S27 in the Supplementary Information) were conducted from March 12 to May 31, 2013, with a total measurement duration of around 27 hours. For each flight, the Falcon-20 aircraft with SPIRIT on board flew in the wake of commercial flights, most of which are on the Paris – North Africa route. This allows the detection of aircraft emissions and the study of the air mass aging of contrails. As shown in Figure 4, most measurements were conducted in the region over Toulouse and the Southwest of France, and several flights reached the Atlantic Ocean and the Mediterranean Sea.

Figure 5 shows the measurements during the 20 March 2013 flight, since they are at the heart of the study, i.e., the emissions of $NO_2$ and CO by aircraft. Sudden increases of $NO_2$ by a few ppbv (from a background level of less than 0.5 ppbv) were observed when chasing aircraft a few minutes in the wake behind them, but no increase of CO as seen here in the middle of ceiling altitude (10.5 km), suggesting complete combustion in the aircraft engines or very rapid dilution process if there are emissions.

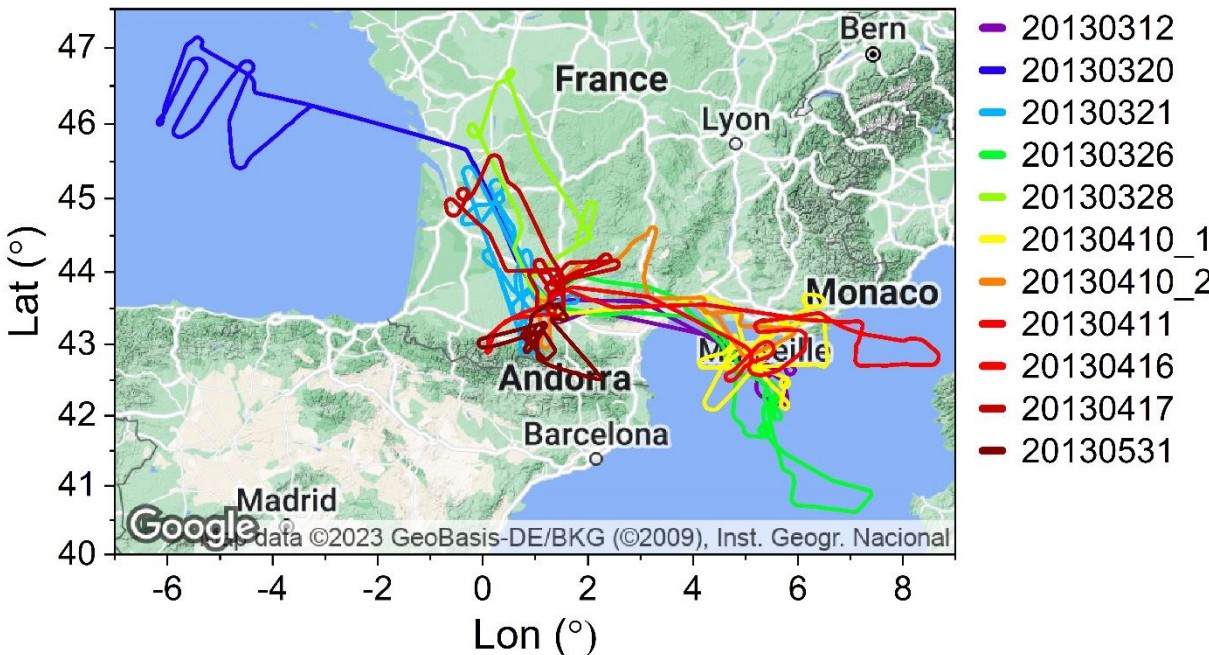

**Figure 4: Flight trajectories during the TC2 2013 campaign. Map copyright: © GoogleMap.**

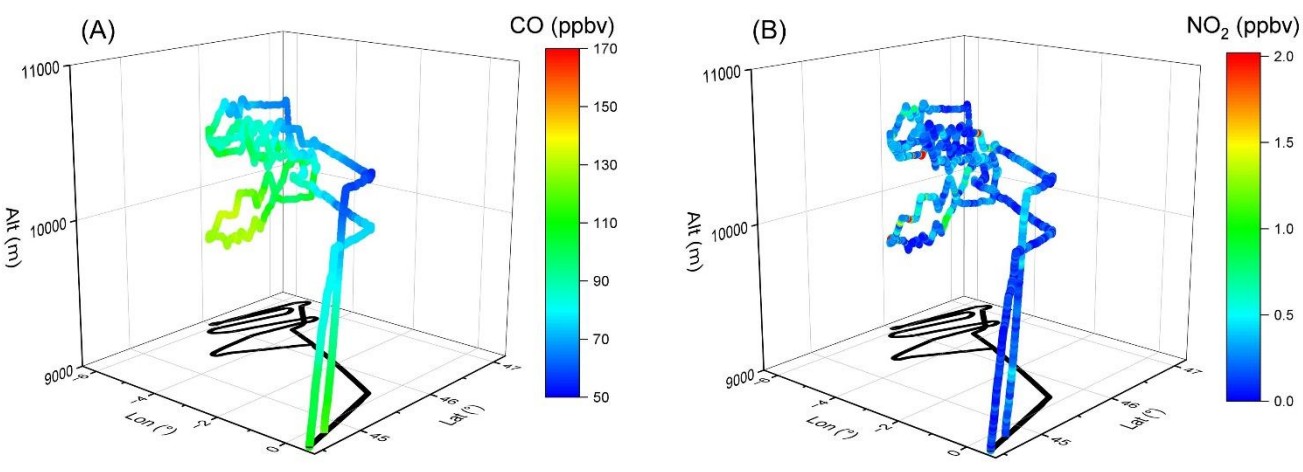

**Figure 5: Flight trajectory colored by CO and NO₂ vmr on flight-TC2_20130320_1. Three NO₂ vmr sudden increases up to more than 2 ppbv (symbolised by red dots) are seen when our research aircraft was inside the contrails emitted by commercial aircraft, when there was no such increase in CO level. Map copyright: © GoogleMap.**

## 3.3 ChemCalInt - France (2014)

The ChemCalInt project is a French initiative part of the European JRA TGOE in EUFAR for the traceability in gas-phase
observations' on-board research aircraft. This project was built to compare the measurements of CO and CH4 by different
airborne instruments in order to ensure their consistency and their performance and to implement adapted calibration
procedures in future campaigns. Figure 6 shows all the flight trajectories during this project. Four flights were conducted in
southern France, generally between/around Toulouse, Clermont-Ferrand, and the Pyrenees Mountains (Figures S28-S31).

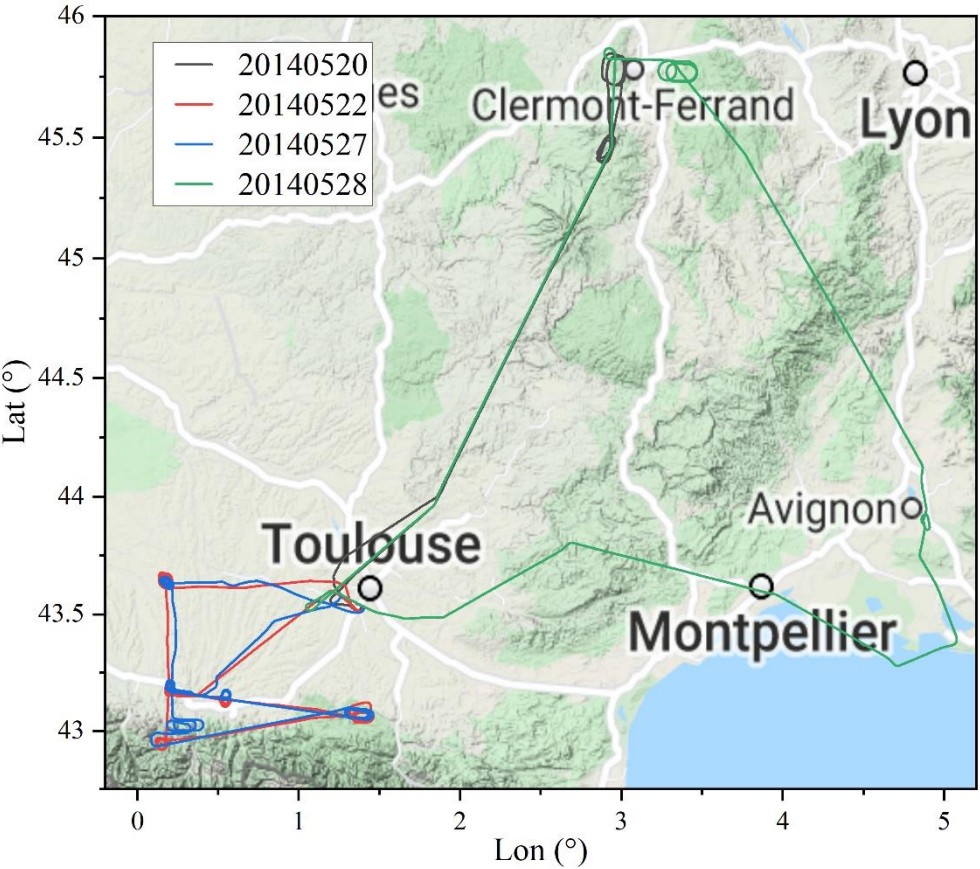

**Figure 6: Flight trajectories during the ChemCalInt 2014 campaign. Map copyright: © GoogleMap.**

In Figure 7, CO measurements along with the flight trajectories on May 22, 2014 are shown since they represent the typical
CO distribution and level in this region. The aircraft flew to the meteorological station of Peyrusse-Vieille, the Centre de
Recherches Atmosphériques de Lannemezan (OMP), the Pyrenees National Park, and the Regional Natural Park of the
Ariegean Pyrenees. Over this natural park, a background measurement of the vertical profile of CO (i.e., < 115 ppbv) was
observed. In general, the measured CO vmr during this flight was lower than 120 ppbv, with an exception over Toulouse. This

is the typical CO level in Southwest France with no big cities and no significant anthropogenic emissions in this region. The vertical homogeneity of CO over the Regional Park is representative of CO distribution over this region as seen in Section 3.9 and Figure 25.

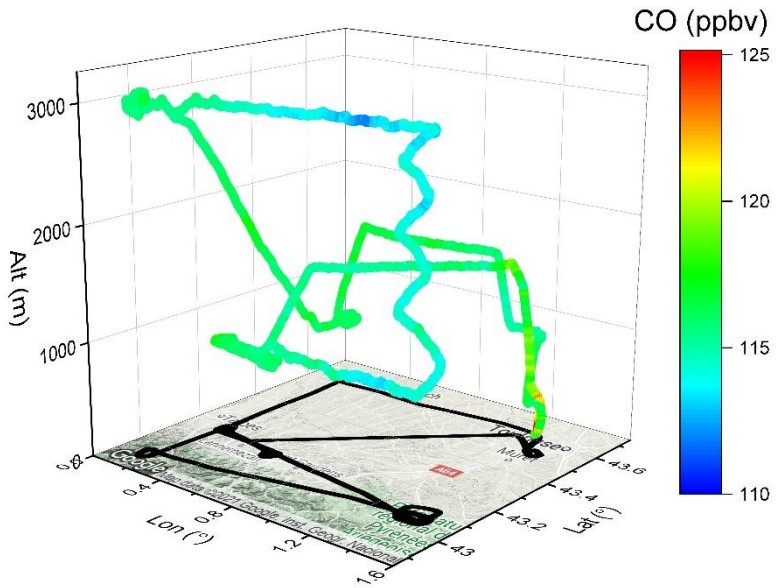

**Figure 7: Flight trajectory colored by CO levels on flight- ChemCalInt_20140522_1. Map copyright: © GoogleMap.**

### 3.4 ChArMEx – GLAM Mediterranean (2014)

Within the frame of the MISTRALS-ChArMEx program (https://programmes.insu.cnrs.fr/mistrals/programmes/charmex/, last access: July 18, 2023), the GLAM experiment (**G**radient in **L**ongitude of **A**tmospheric constituents above the **M**editerranean basin, https://www.safire.fr/content_page/32-campagnes-passees/106-glam.html?lang=fr, last access: July 18, 2023) aims to describe the high-resolution horizontal and vertical distribution of pollutants over the Mediterranean Basin along an East-West axis (Ricaud et al., 2018; Brocchi et al., 2018). As shown in Figure 8, the measurement domain of the 9 flights covers the target region, west from the France-Spain border and east to Cyprus (Figures S32-S40). Measurements were mostly conducted over the sea, with some exceptions for islands including Balearic Island, Sardinia Island, Crete Island, and Cyprus Island, drawing the whole picture of CO profiles over the Mediterranean Basin.

Data obtained in this project can be used to study anthropogenic emissions in the Mediterranean region, for which measurements on August 8, 2014 provide an excellent example. Figure 9 shows measurements during this flight with a measurement area over Cyprus and surrounding regions. High levels of CO up to 130 ppbv were observed in the boundary layer (< 2 km altitude) over the coastal regions of south-eastern Cyprus and the sea on the west of Cyprus, indicating impacts of urban and shipping emissions, respectively.

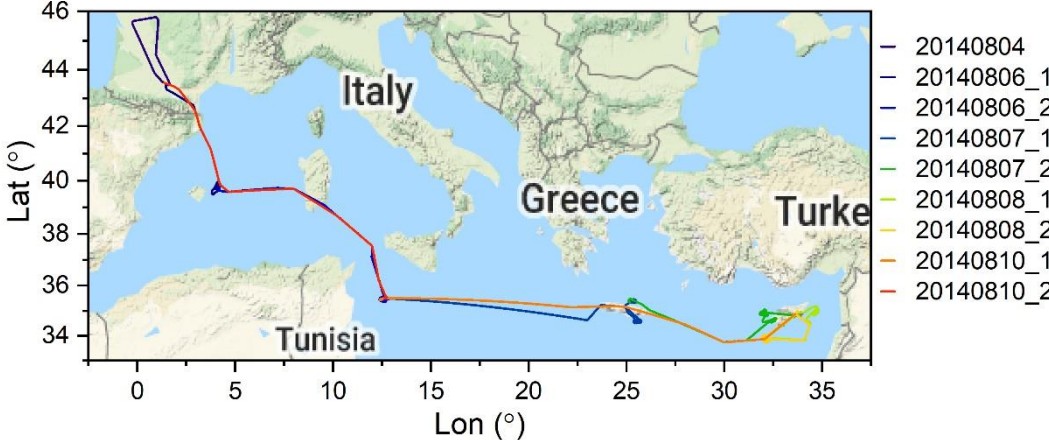

**Figure 8: Flight trajectories during the GLAM-2014 campaign. Map copyright: © GoogleMap.**

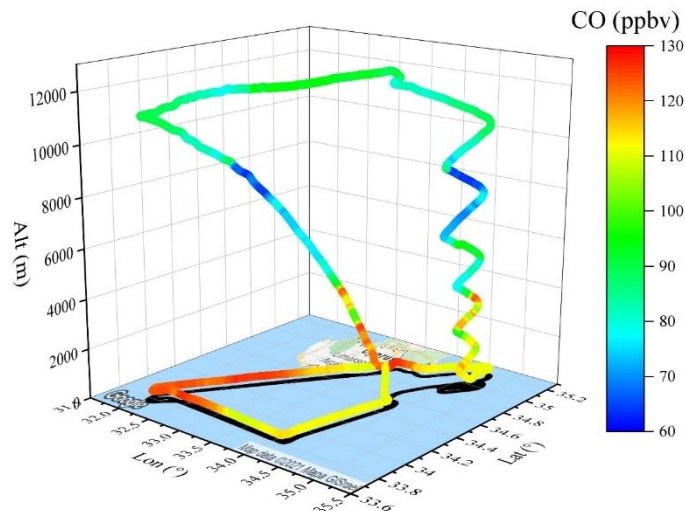

**Figure 9: Flight trajectory colored by CO levels on flight-GLAM_20140808_1.  Map copyright: © GoogleMap.**

### 3.5 Test-ATR-42 (2016)

The Test-ATR-42 project aims to test the performances of SPIRIT, with more intensive calibrations during the flight compared to other field campaigns, as detailed in Catoire et al. (2017). As shown in Figure 10, 17 calibrations with the WMO CO standard of $136.1 \pm 0.4$ ppbv were made during the flight on February 5, 2016. Those calibrations cover both ascending and descending of the aircraft in the altitude range of 0 – 6 km, showing no dependence of the results on temperature or pressure. Each time the sampling inlet is connected to the cylinder containing the WMO CO standard, measurements rapidly (in 10 s) reach a stable

value around 140 ppbv, with a small scattering (< 1 ppbv at 1σ, Figure 11), highlighting the accuracy and reproducibility of SPIRIT in aircraft measurements.

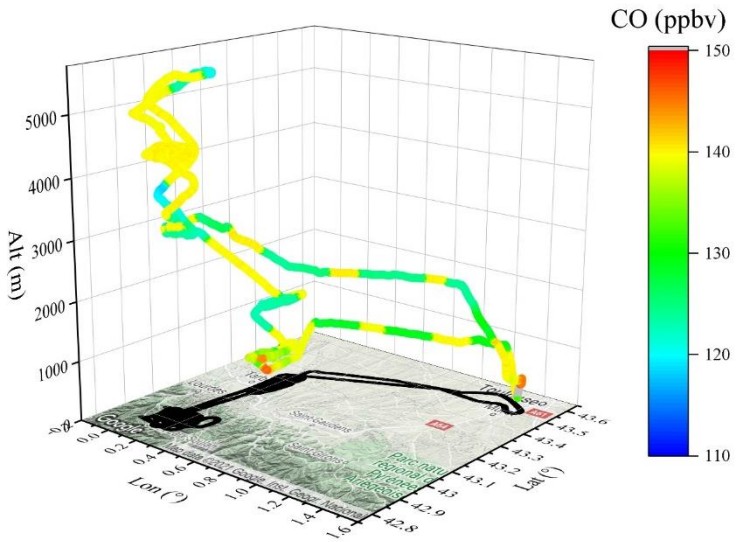

**Figure 10: Flight trajectory colored by CO levels on flight-Test-ATR-42_20160205_1. Map copyright: © GoogleMap.**

220

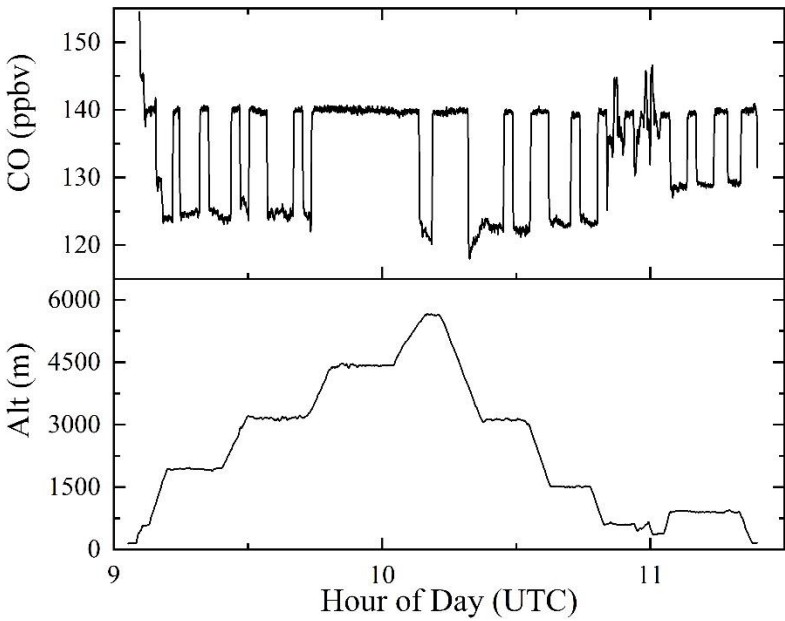

**Figure 11: Time series of CO and altitude on flight- Test-ATR-42_20160502_1.**

## 3.6 DACCIWA – West Africa (2016)

In the summer of 2016, the EU FP7 project DACCIWA (Dynamics-Aerosol-Chemistry-Cloud Interactions in West Africa) was implemented to investigate the atmospheric impacts of anthropogenic and natural emissions in West Africa, where high-quality airborne measurements are quite scarce (Knippertz et al., 2015; Hahn et al., 2022; Taylor et al., 2019; Wu et al., 2021; Formenti et al., 2019; Redemann et al., 2021; Haywood et al., 2021). 14 flights (> 50 h measurements; Figures S42-S55) were conducted in coastal regions as well as the surrounding Atlantic Ocean and continental regions of West Africa (Figure 12). In general, CO levels in West Africa are higher than those in the Mediterranean Basin (Section 3.4).

For example, Figure 13 shows a flight along the coastal lines and over the Gulf of Guinea. This represents a typical measurement as it is impacted by different types of pollutions. High CO values (200 – 300 ppbv) were observed within 2 km above Lomé (Togo), suggesting urban emissions and/or shipping emissions (see ship positions near the Lomé port in Figure S65). Moreover, Brocchi et al. (2019) found that offshore oil rig emissions (e.g., CO, $NO_x$, and aerosol within the boundary layer) also showed an impact on regional air quality. The pollutants emitted above the ocean by ships and oil rigs are thus transported along the West African Monsoon (south-westerly in summer) to the continents (Kniffka et al., 2019; Brocchi et al., 2019), affecting the air quality in those regions. Notably, a pollution plume caused by biomass burning emissions, with aerosols visible live through the aircraft window and higher CO levels (150 – 200 ppbv) than the background (100 – 150 ppbv, Figure 13), was observed at 2.8 – 4.0 km and was captured again at a similar altitude but further from the coast.

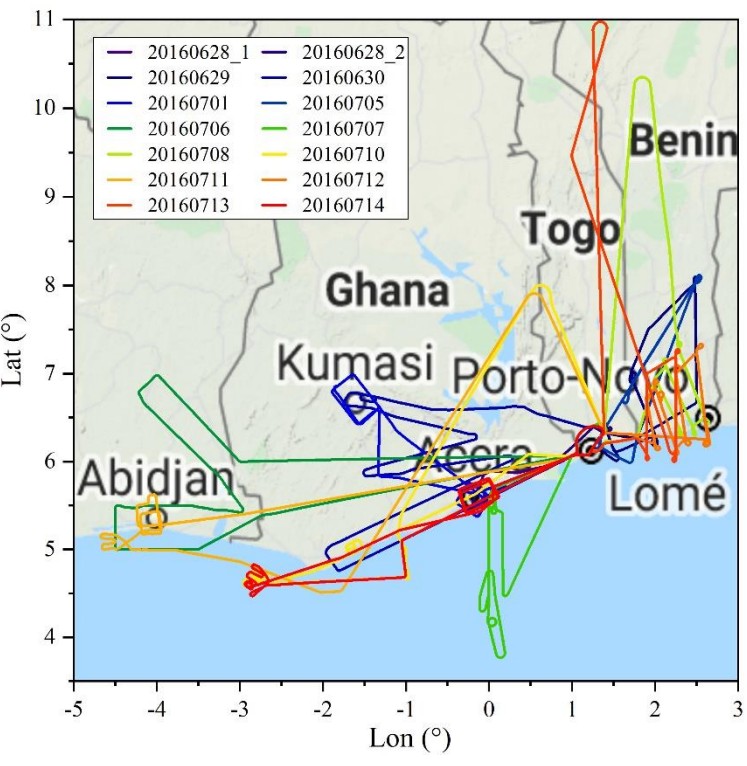

     **Figure 12: Flight trajectories during the DACCIWA 2016 campaign. Map copyright: © GoogleMap.**

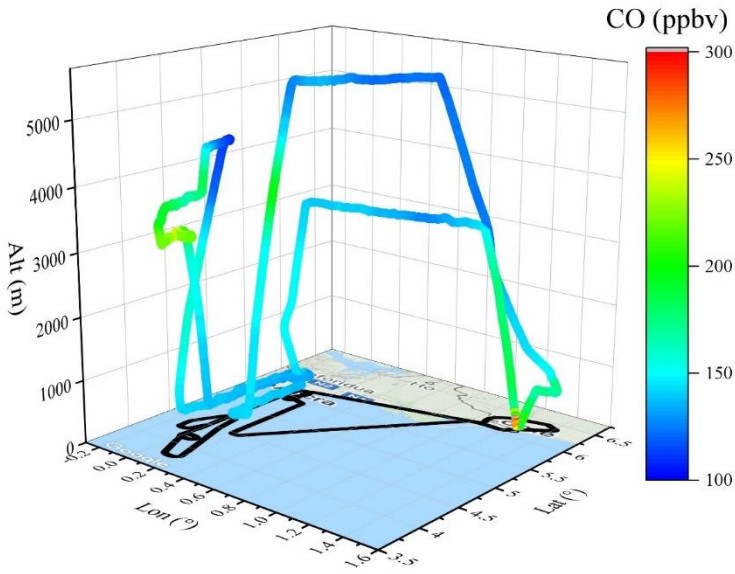

**Figure 13: Flight trajectory colored by CO levels on flight-DACCIWA_20160707_1. Map copyright: © GoogleMap.**

Local emissions were frequently observed during this campaign. For instance, as shown in Figure 14, the flight on July 12, 2016 allowed intensive measurements between Lomé (Togo) and Porto-Novo (Benin). Most measurements over the continent showed higher CO levels than those over the ocean, with several CO peaks up to 200 ppbv occasionally accompanied by peaks of $NO_2$ and $SO_2$ (not shown). Those peaks were observed within the boundary layer, indicating the impact of anthropogenic emissions in this region. These measurements are important for documenting and understanding local air pollution, considering that the increasing population and economy lead to more anthropogenic emissions (Knippertz et al., 2015). Also, aerosol pollution traced by CO levels (> 155 ppbv) is shown to slightly enhance atmospheric cooling by low-level clouds in tropical West Africa by increasing the droplet number concentration and reducing the droplet size in the clouds (Hahn et al., 2023; Taylor et al., 2019).

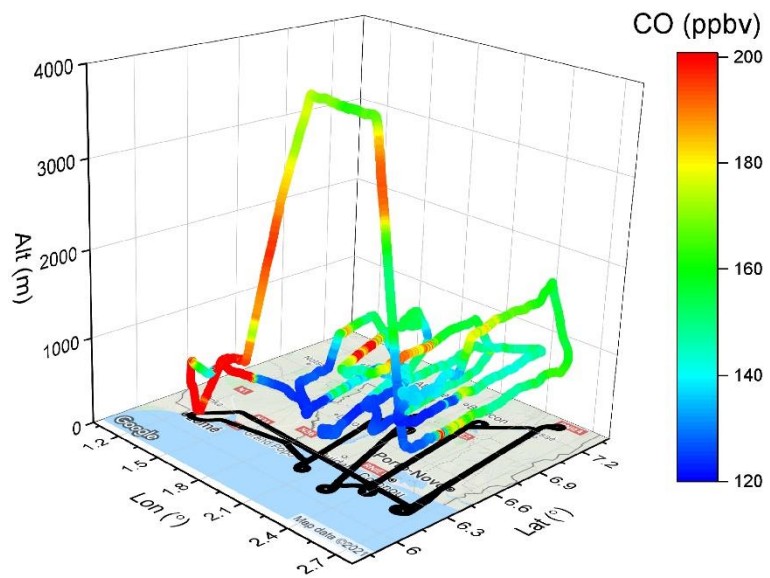

**Figure 14: Flight trajectory colored by CO levels on flight-DACCIWA_20160712_1. Map copyright: © GoogleMap.**

### 3.7 MAGIC (2019)

MAGIC (Monitoring Atmospheric composition and Greenhouse gases through multi-Instrument Campaigns, https://magic.aeris-data.fr/, last access: July 18, 2023) is a long-term CNES-CNRS French project, with two main goals: (i) to better understand the vertical exchange of greenhouse gases along the atmospheric column, in connection with atmospheric transport, sources and sinks of the gases at the surface and in the atmosphere; (ii) to contribute to the validation of satellite products of IASI-MetOpC (CNES), Tropomi-Sentinel 5P (ESA), GOSAT 2 (JAXA), and OCO-2 (NASA), and to prepare the

validation of space missions MicroCarb, IASI-NG (CNES), and Merlin (CNES-DLR) dedicated to the monitoring of greenhouse gases and other species. In June 2019, the MAGIC campaign was conducted in France, with three flights over the Toulouse-La Rochelle-Orléans-Clermont-Ferrand region (Figure 15 and Figures S56-S58 for 3D trajectories with CO vmr).

Similar to the Test-ATR project (Section 3.5), intensive in-flight calibrations were conducted, which led to the plots of CO versus flight trajectories (Figures S56 – S58). Figure 16 shows the measurements during calibration for the two flights on June

18, 2019. The eight calibration processes cover ascending, descending, and cruise periods in the altitude range of 0 – 12 km. This results in an agreement between the WMO CO_X2014A standard and the SPIRIT instrument within $3.2 \pm 0.6$ ppbv ($1\sigma$) out of 149.0 ppbv, confirming the accuracy and stability of SPIRIT for aircraft measurements.

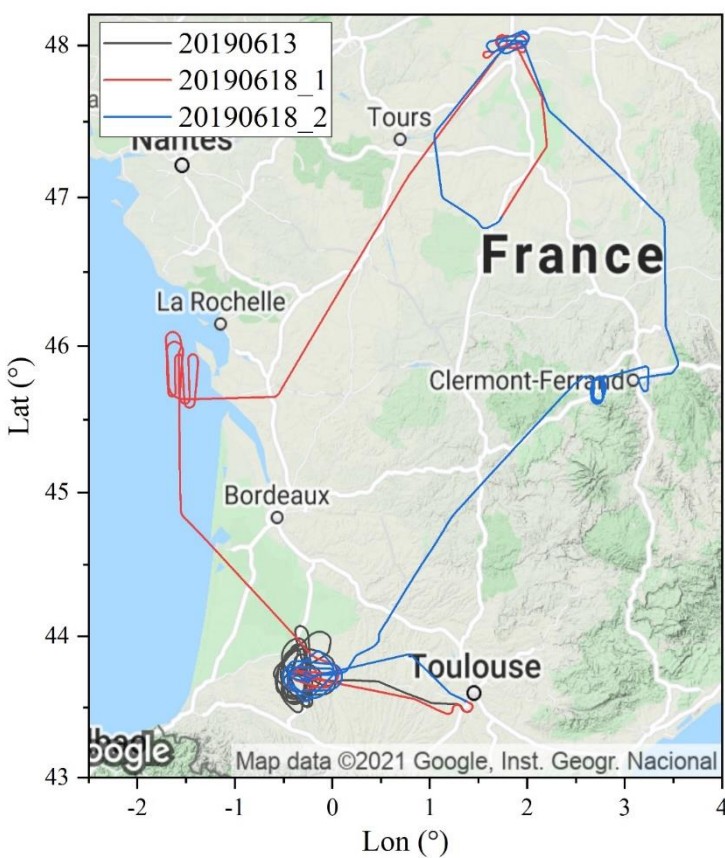

**Figure 15: Flight trajectories during the MAGIC 2019 campaign. Map copyright: © GoogleMap.**

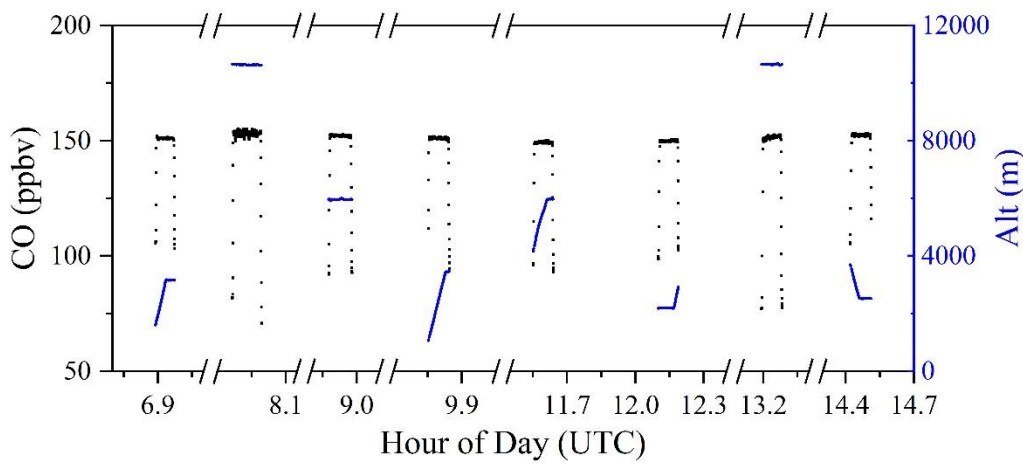

**Figure 16: Time series of CO values (black dot) and altitude (blue dot) during calibration on flight-MAGIC_20190618_1 and MAGIC_20190618_2.**

## 3.8 MAGIC (2021)

In the continuation of the MAGIC 2019 project above, exploring high latitude regions is of interest since these ones are
275 warming faster than the average, as a result of anthropogenic, natural emissions and transport of pollution. However, airborne
measurements are scarce in high-latitude regions, which limits the understanding of the horizontal and vertical distribution of
pollution and causes problems in validating satellite performance in those regions. In August 2021, the international MAGIC-
2021 campaign (https://magic.aeris-data.fr/, last access: July 18, 2023) took place in Kiruna (67.9ºN, 21.1°E), Sweden. As
shown in Figure 17, the six flights mainly performed measurements over northern Sweden and northern Finland, with one
flight reaching the Norwegian Sea (Figures S59-S64).

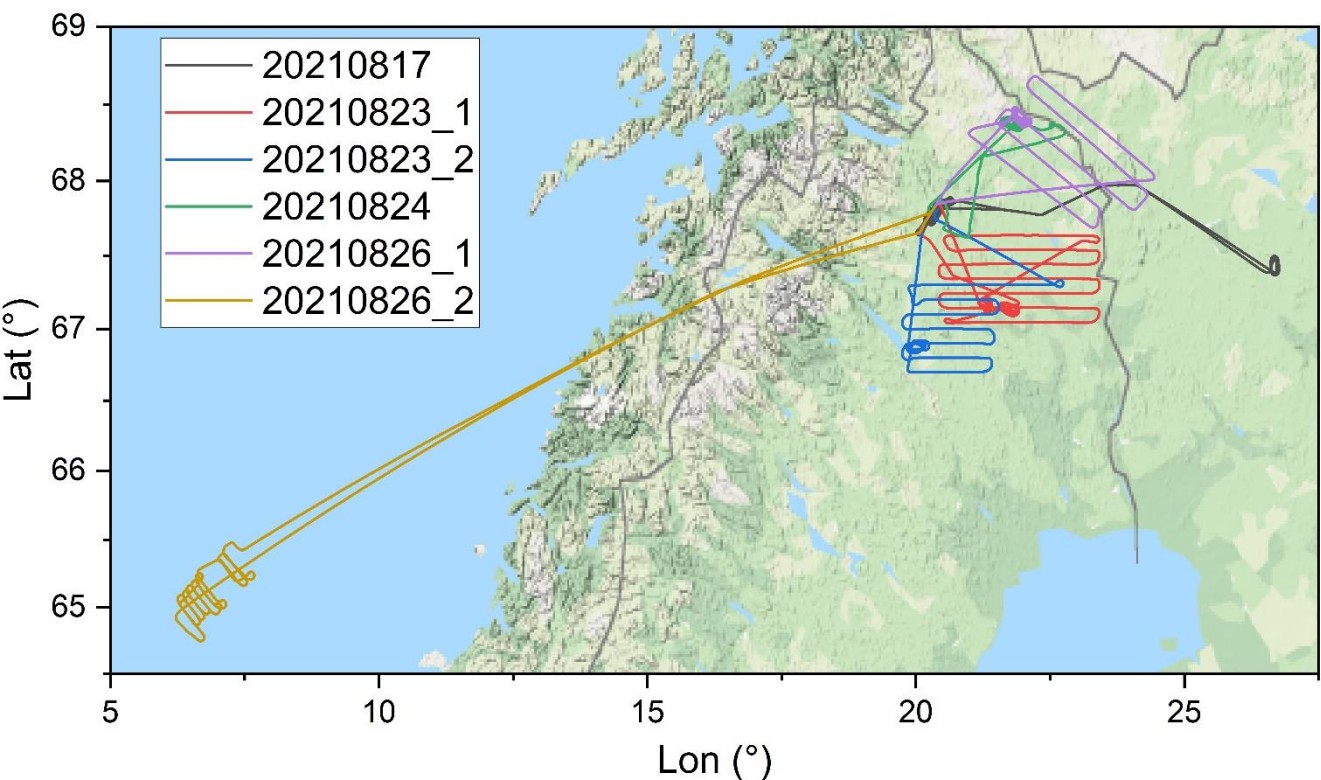

**Figure 17: Flight trajectories during the MAGIC 2021 campaign. Map copyright: © Google Map.**

On the morning of August 23, 2021, a flight was designed to provide measurements for the validation of satellite products
aforementioned in Section 3.7. Intensive vertical and horizontal measurements were carried out, synchronized with several
satellite overpasses. As shown in Figure 18, the aircraft spiralled up, achieving two vertical profiles of measurements within
an altitude range of 0 – 7.5 km asl. At the maximum altitude, the aircraft descended to 4.5 km asl and conducted intensive
horizontal measurements in the region of 67.0 – 67.6 ºN and 20.0 – 23.5ºE. CO vmr was mostly around 150 – 250 ppbv in this
region. After that, it spiralled up again to 7.5 km asl, followed by spiraling down to the ground level before landing back at

Kiruna airport. CO was mostly around 150 ppbv within the boundary layer of ~2 km asl. However, it increased with altitude above the boundary layer (peak of 285 ppbv at ~5.5 km asl, Figure 29), likely due to long-range transported fire pollution (study in progress).

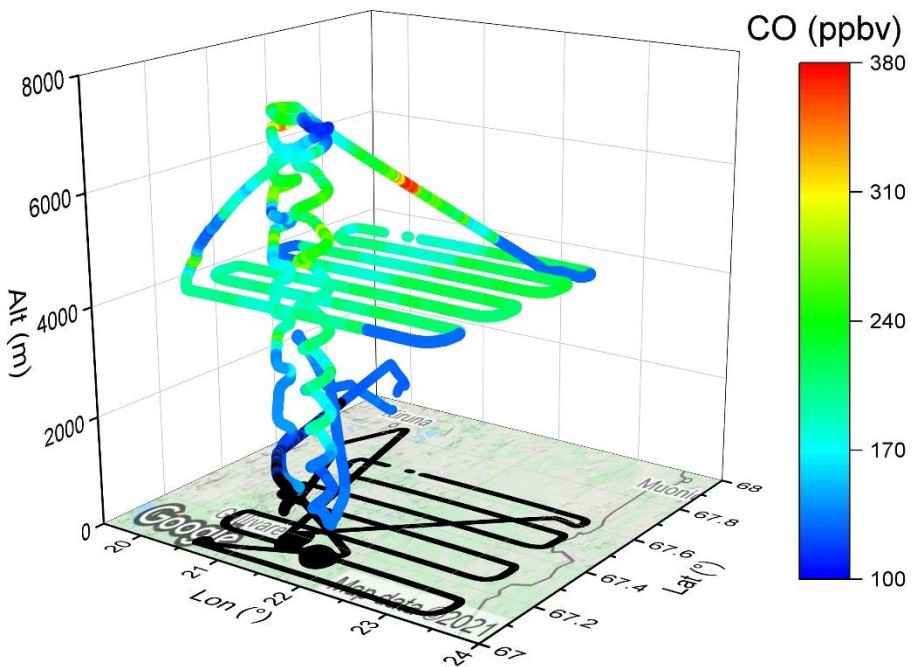

**Figure 18: Flight trajectory colored by CO levels on flight-MAGIC_20210823_1. Map copyright: © Google Map.**

In addition to long-range transport, local emissions from fires were also observed. Measurements on August 26 showed the impact of two pollution sources. At 9:30 (half an hour after starting the 3.6-hour flight-20210826_1) on August 26, a fire caused by a rocket engine test occurred at the Swedish Space Corporation Esrange base (67.9 ºN, 21.1 ºE), which burned parts of the launching facility for sounding rockets and parts of the nearby buildings (red star in Figure 19). The plume of this fire was captured by our measurements. As shown in Figure 19, the aircraft took off at Kiruna airport at around 9:05 a.m. (local time). Around 5 min after the take-off, high CO peaks of more than 200 ppbv were observed at an altitude of 2000 – 2750 m asl, which is nearly above the location of the fire at the Esrange base. The plume was also frequently observed at higher altitudes, i.e., four CO peaks of above 200 ppbv were observed at around 5600 m asl and distributed on a straight line, indicating the widespread of the plume.

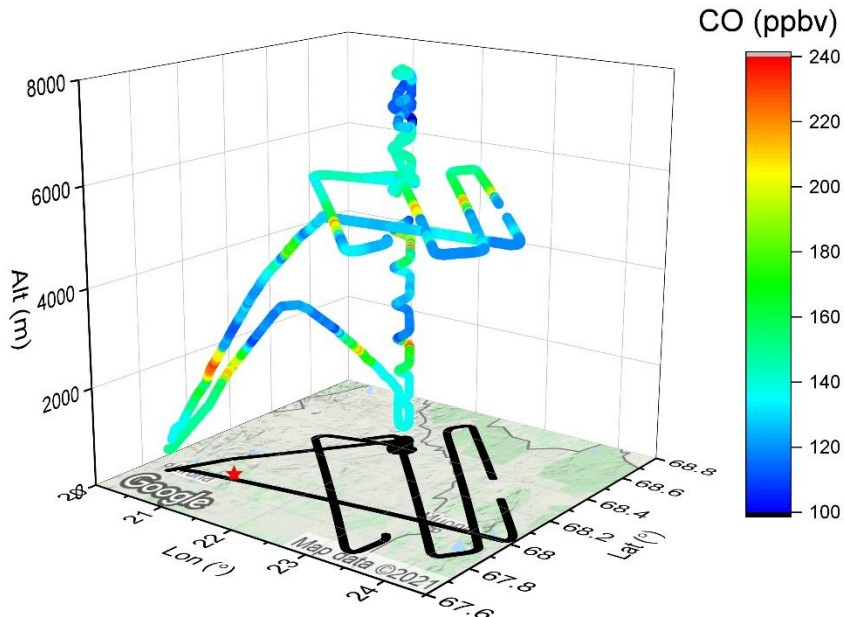

**Figure 19: Flight trajectory colored by CO levels on flight-MAGIC_20210826_1. The red star represents the location of the fire at Esrange. Map copyright: © Google Map**

### 3.9 Intercontinental Measurements

Two intercontinental measurements were achieved during SHIVA and DACCIWA projects. Aircraft travelled from Germany to Malaysia in South East Asia in November 2011 and to Ghana in West Africa in June 2016. Except for the landing and take-off, the measurement height was typically constant at a typical cruise altitude of ~10 km asl. This could be used for assessing the performance of chemical transport models and validating satellite observations on a global scale.

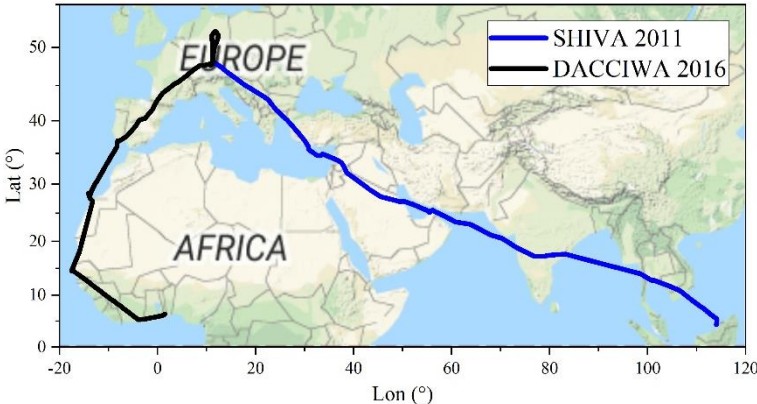

**Figure 20: Trajectories of the two inter-continent flights during the SHIVA 2011 and DACCIWA 2016 campaigns. Map copyright: © Google Map.**

### 3.9.1 Europe – Asia

In the framework of SHIVA, an intercontinental measurement was conducted from Munich (Germany) to Miri (Malaysia), with four stops at Larnaca (Cyprus), Dubai (United Arab Emirates), Hyderabad (India), and Pattaya U-Tapao (Thailand), respectively (Figure 21). Except for CO peaks observed after the take-off, the measured CO was typically lower than 110 ppbv. Higher CO levels (> 130 ppbv) were observed in the boundary layer over Hyderabad and U-Tapao, indicating the impacts of local anthropogenic emissions. The pollution of the boundary layer above these last two cities is clearly visible, with CO levels

well above 200 ppbv. Also, before landing at Miri, a polluted plume, with significant CO enhancements from around 80 to 200 ppbv, was observed at 10680 m asl. The width of the measured plume was about 630 km, indicating a large emission event and a wide spread. The observed plume may originate from events with intensive emissions, such as wildfires and convective transport (Krysztofiak et al., 2018; Hamer et al., 2021).

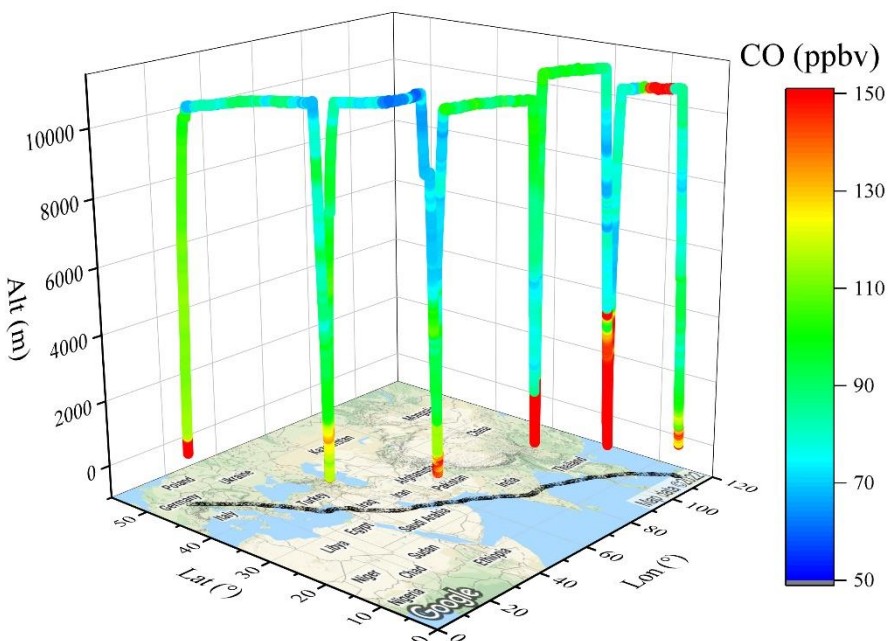

**Figure 21: Flight trajectory colored by CO levels during the inter-continent flight during the SHIVA 2011 campaign. Map copyright: © Google Map.**

### 3.9.2 Europe – Africa

During the DACCIWA project, the SPIRIT instrument was on board the DLR Falcon-20 for measurements during the transit flights from Germany to Togo. There were three stops: Faro (Portugal), Fuerteventura (Canary Island, Spain), and Dakar

(Senegal), which allows measurements across West Europe and along the coastal lines of North and West Africa. CO measured over West Europe and North Africa is typically lower than 100 ppbv. However, higher CO levels were observed over West

Africa in the boundary layer and the upper troposphere. For example, before landing at Lomé (Togo), polluted plumes with CO up to 150 ppbv were observed in the altitude range of 20 – 4700 m asl, suggesting strong impacts of regional anthropogenic emissions.

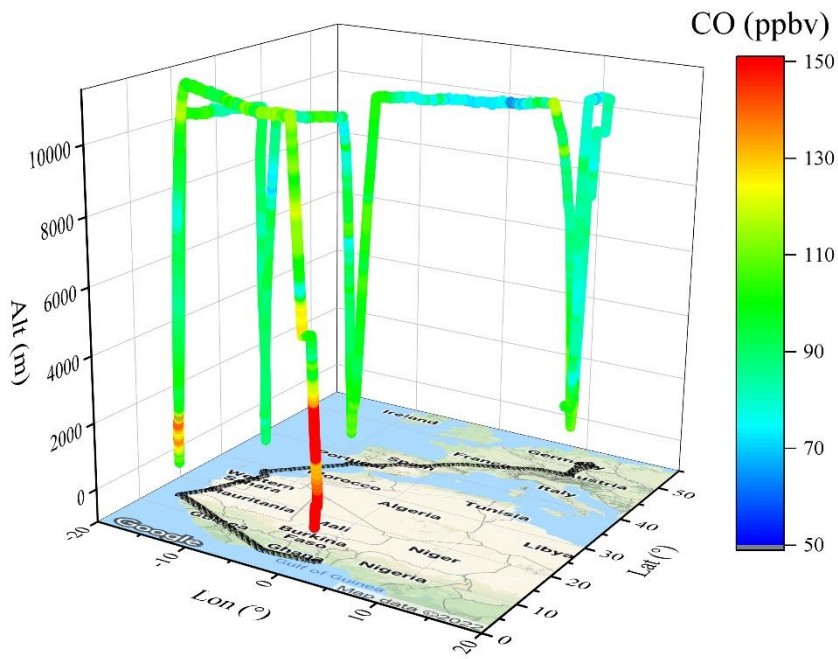

**Figure 22: Flight trajectory colored by CO levels during the intercontinental flight during the DACCIWA 2016 campaign. Map copyright: © Google Map.**

### 3.10 Vertical Profiles of CO in the Different Regions

Figure 23 – 29 show the vertical profiles of CO observed during the campaigns. In most cases, CO levels decrease with altitude within the boundary layer (~1-2 km asl) from 100 – 200 ppbv (depending on the region) to lower values in the free troposphere, characterized as such, indicating a strong impact of anthropogenic emissions. Similar observations are made in Alaska where a nearly constant CO value was observed in an altitude range of 0 – 2 km asl (Spackman et al., 2010). However, surface CO vmr in West Africa (Figure 27) are generally in the range of 200 – 300 ppbv. They are higher than in other regions. Indeed, in Europe and the whole Mediterranean Basin (including TC2-2013, ChemCallnt-2014, and GLAM-2014), most surface CO levels are lower than 150 ppbv. In southern Asia, surface CO is typically at a similar level to Europe, with exceptions reaching more than 200 ppbv.

Unlike other regions, the CO vertical profile measured in high-latitude regions (Kiruna, Sweden) during the MAGIC-2021 campaign increases with altitude, with CO levels of up to 300 ppbv in the altitude range of 5000 – 7000 m asl (Figure 28). CO levels during summer seem similar or higher than inland Europe although a lower anthropogenic emission is expected,

suggesting a large contribution from the transport of plumes from wildfires at high latitude regions. Indeed, assimilated TROPOMI satellite observations and CAMS model simulations reported large CO emissions from wildfires over North America and Russia in July and August 2021, which led to high CO column concentrations in the Arctic region.

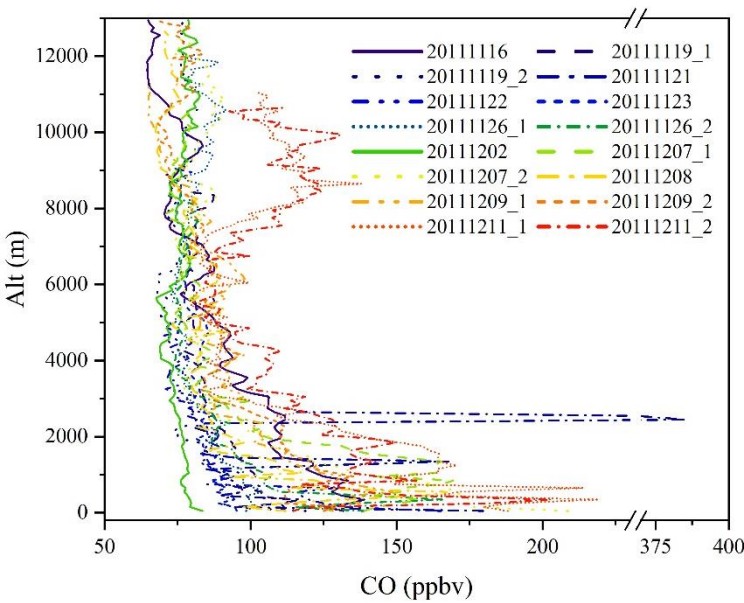

**Figure 23: Vertical profiles of CO of all flights during the SHIVA (2011) campaign in Southern Asia. Vertical average bin: 100 m.**

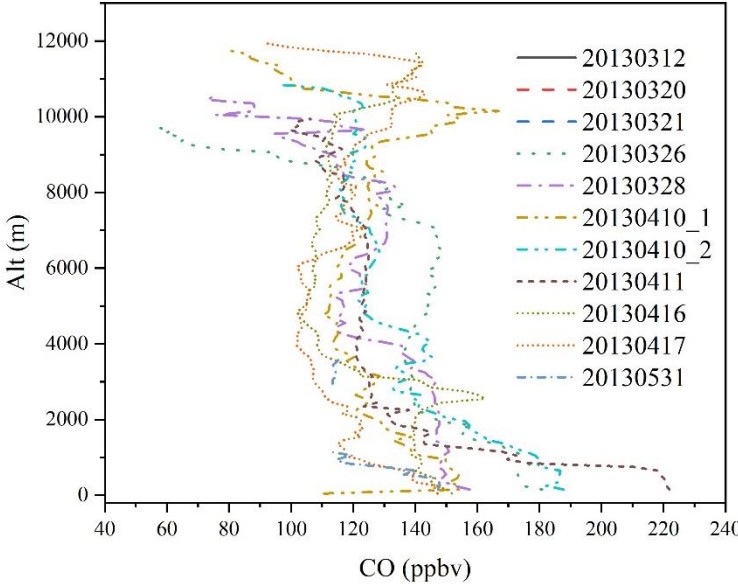


**Figure 24: Vertical profiles of CO of all flights during the TC2 (2013) campaign in Southwestern France. Vertical average bin: 100 m.**

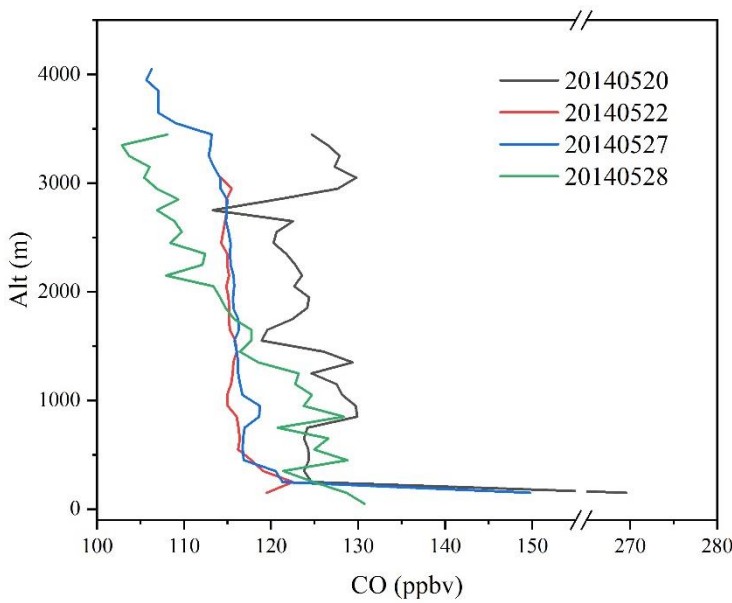

**Figure 25: Vertical profiles of CO of all flights during the ChemCallnt (2014) campaign in Southern France. Vertical average bin: 100 m.**

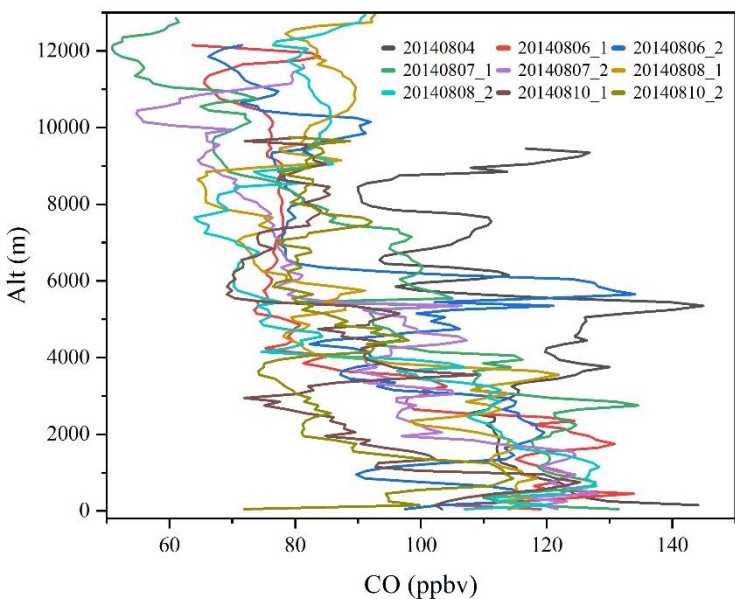

**Figure 26: Vertical profiles of CO of all flights during the GLAM (2014) campaign over the Mediteraneanean Basin. Vertical average bin: 100 m.**

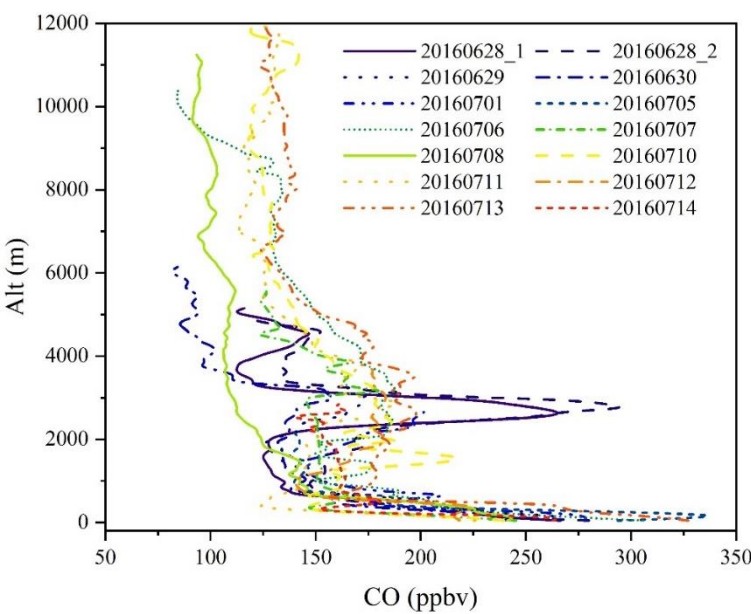

**Figure 27: Vertical profiles of CO of all flights during the DACCIWA (2016) campaign in West Africa. Vertical average bin: 100 m.**

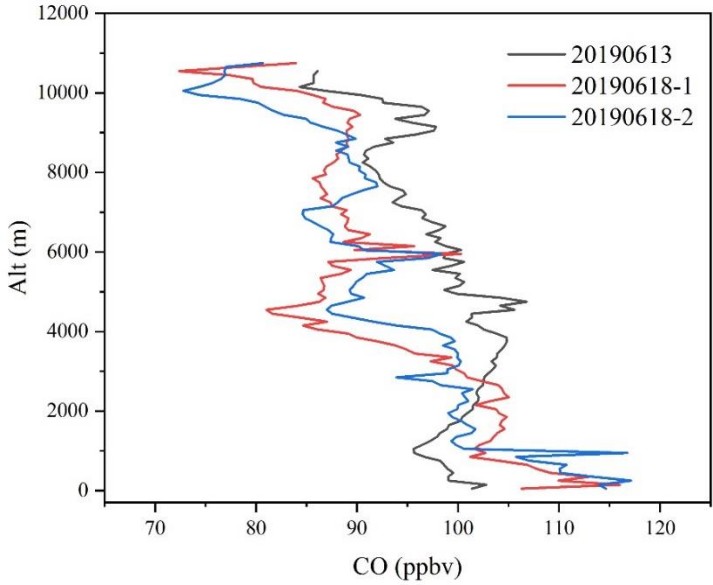

**Figure 28: Vertical profiles of CO of all flights during the MAGIC (2019) campaign in France. Vertical average bin: 100 m.**


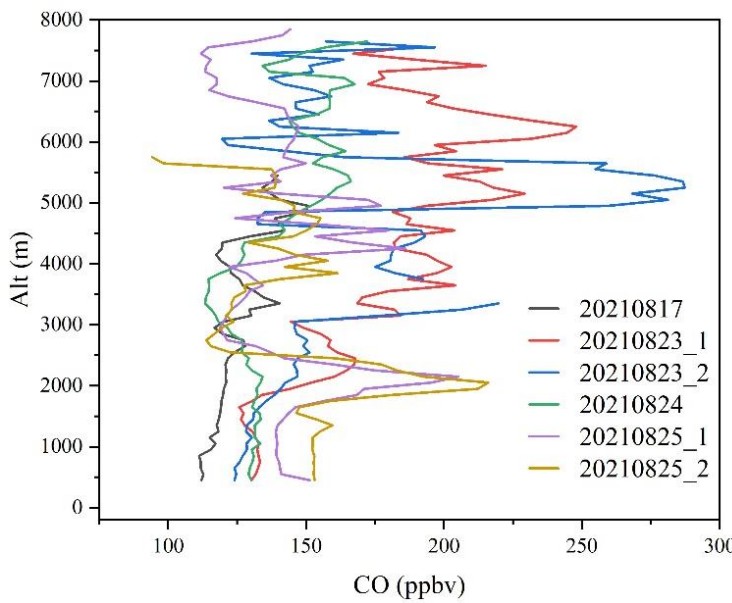

**Figure 29: Vertical profiles of CO of all flights during the MAGIC (2021) campaign in Northern Sweden. Vertical average bin: 100 m.**

## 4 Data Availability

All the data used in this study are publicly available on the AERIS database (Catoire et al., 2023: https://doi.org/10.25326/440). Any questions concerning this current database or the scheduled SPIRIT aircraft measurements in the future are welcomed by contacting the corresponding author. Furthermore, there are also CO measurements on aircraft and balloon platforms referenced in AERIS database (https://www.aeris-data.fr/en/catalogue-en/), such as in-situ measurements during StraPolÉté in 2009 (Krysztofiak et al., 2012) and off-line AirCore measurements in 2017 (Hooghiem et al., 2020).

## 5 Code Availability

The 3D aircraft trajectories were plotted by OriginPro 2021 (https://www.originlab.com/, last access: July 18, 2023, license needed) and the background maps were inputted by a build-in application (Google Map Import, created by the OriginLab Technical Support, available at https://www.originlab.com/fileExchange/details.aspx?fid=344, last access: July 18, 2023).

## 6 Summary and Conclusion

Thanks to the development of the SPIRIT instrument, accurate airborne CO measurements (0.3 ppbv precision at 1σ and overall uncertainty < 4 ppbv) were achieved during the past decade (2011 – 2021). This database describes all aircraft CO

measurements by SPIRIT during this period. More than 200 h of airborne measurements were conducted in Europe, South Asia, and West Africa. The measurement domain covers a wide area, including tropical, northern mid-latitude, and northern high-latitude regions. This database also includes two unique intercontinental measurements, one from Germany to Ghana via the West African coast, and the other one from Germany to Malaysia.

Generally, surface CO levels in Southern Asia, Europe, and the whole Mediterranean region Basin are at a similar level, namely with mixing ratios < 150 ppbv, which are lower than that the coastal region of West Africa (200-300 ppbv). Moreover, CO levels decrease with altitude for most measurements except at high latitude regions where CO increases with altitude. This database can be of particular interest for studying anthropogenic emissions (DACCIWA-2016), aviation emissions (TC2-2013), wildfire emissions (SHIVA-2011, DACCIWA-2016), shipping and offshore oil rig emissions (GLAM-2014, DACCIWA-2016), convective plumes (SHIVA-2011), urban background (ChemCalInt-2014, MAGIC-2019), and long-distance transport (GLAM-2014, MAGIC-2021). This database helps to understand such events and to constrain and improve relevant model approaches. For instance, based on SHIVA data, it is found that the convective system can significantly affect the composition of the upper troposphere (Krysztofiak et al., 2018; Hamer et al., 2021). Siberia forest fire could affect the atmosphere over the Mediterranean Basin through long-distance transport, which has been confirmed by measurements during the GLAM-2014 project with a combination of a trajectory model and a chemistry-transport model (Brocchi et al., 2018). Brocchi et al. (2019) analyzed DACCIWA measurements and found that offshore oil rigs could contribute to deteriorating the air quality in coastal regions of West Africa. Moreover, with the emerging utilization of satellite images, the importance of calibration of those measurements at a regional and/or a global scale becomes more and more significant, revealing the importance of the SPIRIT database. For example, CO measurements during the DACCIWA-2016 project will be used to validate satellite product from Measurement of Pollution in the Troposphere (MOPITT).

**7 Author Contribution**

C.X., G.K., and V.C. lead this study. All authors participated in- the development, data acquisition, and/or maintenance of SPIRIT during at least one field campaign. All authors commented on and approved this manuscript.

**8 Competing Interests**

The authors declare that they have no competing financial interests.

**9 Acknowledgments**

We are grateful to the LPC2E colleagues Gilles Chalumeau, Thierry Vincent, Kevin Le Letty, Olivier Chevillon, and Anne-Laure Pelé for the mechanical, electronic, optical development and aircraft campaign support and data analysis of the SPIRIT

instrument, and the Master students Guillaume Robelet and Abdelmalek Bakha (University of Orléans) for their help with the data retrieval and visualization. We thank Hans Schlager (DLR) for facilitating access to SHIVA and DACCIWA projects and the associated DLR Falcon-20 aircraft. We thank SAFIRE (CNRS, Météo-France, CNES) for flying research aircraft in excellent conditions.

**Funding**: This work was supported by the PIVOTS project provided by the Region Centre − Val de Loire (ARD 2020 program and CPER 2015 − 2020) and the Labex VOLTAIRE project (ANR-10-LABX-100-01, managed by the University of Orléans). Funding also came from the projects SHIVA (226224-FP7-ENV-2008-1), EUFAR2 TransNational Access, DACCIWA (Grant Agreement N°603502), ChArMEx-MISTRALS, ChemCalInt French initiative and MAGIC (CNRS-INSU, CNES, ADEME, Météo-France, CEA).

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
