# Peer review of "A Database of Aircraft Measurements of Carbon Monoxide (CO) with High Temporal and Spatial Resolution during 2011 – 2021"

_Earth System Science Data, 2023_

## Referee Comment (RC1)

Review of

**A Database of Aircraft Measurements of Carbon Monoxide (CO) with High Temporal and Spatial Resolution during 2011 – 2021**

by Xue et al.

The manuscript presents a dataset of high-quality, airborne carbon monoxide measurements. The dataset is based on seven aircraft campaigns and two intercontinental flights with five campaigns over Europe and two campaigns over Asia and Africa, respectively. While the data set is relatively small and limited in terms of temporal and spatial coverage, it is of interest for studies of atmospheric transport and chemistry that require high-resolution and precise measurements. I recommend publication after the following comments have been addressed.

**Major comments:**

The presentation of the data needs to be improved. At the moment the manuscript includes a large number of figures of different quality and resolution. Presentation of all flight tracks in individual figures and in one figure combined seems not necessary. In addition, the presentation of the flight tracks on top of the Google maps shows poor quality for some of the campaigns with low resolution and strangely enlarged or stretched fonts. My suggestion would be to remove the individual campaign flight plots and to either have one very much improved version of Figure 22 or to replace Figure 22 with a scientific style map plot that shows the individual campaigns.

The 3D flight track plots (color-coded by the CO measurements) are interesting but need to be improved and made consistent. Some are missing the vertical axis. Some have black surface lines for the flight tracks, while others don't.

Please make it clearer (one to two sentences in each case is fine) why a particular flight is shown for a campaign by explaining the motivation to present this example.

The manuscript needs an improved conclusion and summary section. It would be valuable to highlight which campaign is typical for particular conditions discussed in the manuscript (anthropogenic emissions, clean background air, ship emissions, convection, wildfires etc.). Please also explain here which campaigns have already been used for either model evaluations or comparisons to satellite measurements.

It would be helpful for interested readers to also point out sources of other airborne, high quality CO measurements (e.g., US campaigns, balloon-borne measurements etc.).

**Minor comments**

Line 37-39: Please rewrite sentence, as it is hard to understand.

Line 54: Not sure what 'atmospheric dynamics … could be achieved' is supposed to mean.

Line 63: What target regions are meant here? Is this supposed to be the upper troposphere in general or some very specific regions?

---

## Author Comment (AC1)

**Comments from Reviewer #1 and Response**

The manuscript presents a dataset of high-quality, airborne carbon monoxide measurements. The dataset is based on seven aircraft campaigns and two intercontinental flights with five campaigns over Europe and two campaigns over Asia and Africa, respectively. While the data set is relatively small and limited in terms of temporal and spatial coverage, it is of interest for studies of atmospheric transport and chemistry that require high-resolution and precise measurements. I recommend publication after the following comments have been addressed.

Our response: Thank you for your valuable feedback, which contributes to improve the overall quality of our manuscript. Please see below the point-to-point response, with your Comments in black, our Responses in blue and Changes in red.

All modified Figures are in the Supplement pdf if not clearly visible in the online Reply.

**Major comments:**

1. The presentation of the data needs to be improved. At the moment the manuscript includes a large number of figures of different quality and resolution. Presentation of all flight tracks in individual figures and in one figure combined seems not necessary. In addition, the presentation of the flight tracks on top of the Google maps shows poor quality for some of the campaigns with low resolution and strangely enlarged or stretched fonts. My suggestion would be to remove the individual campaign flight plots and to either have one very much improved version of Figure 22 or to replace Figure 22 with a scientific style map plot that shows the individual campaigns.

Thank you for the suggestion. First, the individual campaign flight plots are removed from Figure 22 (now labelled Fig. 1). Second, we added bar plots of average CO levels in the boundary layer, lower free troposphere and upper troposphere for each project to enrich the information in Figure 1. The following sentence was added accordingly, line 115: "In general, the CO level depends on the pollution, with the highest values in the boundary layers ($< 2$ km altitude) of areas influenced by anthropogenic activities. The upper troposphere ($> 6$ km altitude) is cleaner (except for MAGIC 2021 high latitude area influenced by long range transport: see more detail in Sections 3.8 and 3.10)."

[Figure]

**Figure 22. Locations of the main airport used for each project. Map copyright: © GoogleMap. The same airport in Toulouse was used for TC2-2013, ChemCallnt-2014, and Test ATR-2016, then the overlapped points are offset and plotted horizontally. Lampedusa E Linosa airport is selected for GLAM-2014 as it is at the center of the measurement area. Barplots show the average CO levels in the boundary layer (0-2 km), lower free troposphere (2-6 km) and upper free troposphere (>6 km) for each project.**

2. The 3D flight track plots (color-coded by the CO measurements) are interesting but need to be improved and made consistent. Some are missing the vertical axis. Some have black surface lines for the flight tracks, while others don't.

We checked all the 3D flight track plots. We modified the Figures 4, 17, 18, S9, S12, S18, S26, S43, S60, S61, S62, S63, and S64, and axes of Figures S13 and S59, by adding or improving the projection of the trajectory.

3. Please make it clearer (one to two sentences in each case is fine) why a particular flight is shown for a campaign by explaining the motivation to present this example.

We added the following explanations for the example flights.

**SHIVA-2011 project:**

"**Erreur ! Source du renvoi introuvable.** exhibits the trajectory of the first scientific flight on 19 November 2011, which gives an insight of the CO level in the boundary layer and free troposphere of the

Borneo island. The aircraft took off from Miri airport (Malaysia), gradually increasing its altitude to 8400 m asl. It flew along the coastal line of the Sarawak state (Malaysia) and reduced its flying height before reaching the southwest of the Sarawak state. Then, after conducting horizontal measurements over the ocean boundary layer, the aircraft increased its flying height and flew back to Miri airport. After about 2.4 hours of measurements, the aircraft landed at Miri airport. During the flight, the observed CO reached 140 ppbv in the boundary layer (0-2 km) and was lower than 100 ppbv in the lower free troposphere (2-6 km), but relatively stable, around 100 ppbv at ~8 km."

**TC2-2013 project:**

The choice of the March 20, 2023 flight is already justified in the text: "since the measurements are at the heart of the study, i.e., the emissions of $NO_2$ and CO by aircraft." Moreover, Figure 5 was modified to also show $NO_2$ variations, with the legend below modified accordingly:

[Figure]

**Figure 1: Flight trajectory colored by CO and $NO_2$ vmr on flight-TC2_20130320_1. Three $NO_2$ vmr sudden increases up to more than 2 ppbv (symbolised by red dots) are seen when our research aircraft was inside the contrails emitted by commercial aircraft, whereas there was no such increase in CO level. Map copyright: © GoogleMap.**

**Chem CalInt-2014 project:**

"In **Erreur ! Source du renvoi introuvable.**, CO measurements along with the flight trajectories on May 22, 2014 are shown since they represent the typical CO distribution and level in this region."

**GLAM-2014 project:**

"Data obtained in this project can be used to study anthropogenic emissions in the Mediterranean region, for which measurements on August 8, 2014 provide an excellent example. **Erreur ! Source du renvoi introuvable.** shows measurements during this flight with a measurement area over Cyprus and surrounding regions. High levels of CO up to 130 ppbv were observed in the boundary layer (< 2 km altitude) over the coastal regions of south-eastern Cyprus and the sea on the west of Cyprus, indicating impacts of urban and shipping emissions, respectively."

**Test-ATR 2016 project:**

There is only one flight during this project designed for testing the performance of the SPIRIT instrument.

**DACCIWA-2016 project:**

The second paragraph of this section has been reworded as follows, also for reasons of English style:

"For example, **Erreur ! Source du renvoi introuvable.** shows a flight along the coastal lines and over the Gulf of Guinea. This represents a typical measurement as it is impacted by different types of pollutions. High CO values (200 – 300 ppbv) were observed within 2 km above Lomé (Togo), suggesting urban emissions and/or shipping emissions (see ship positions near the Lomé port in Figure S65). Moreover, Brocchi et al. (2019) found that offshore oil rig emissions (e.g., CO, $NO_x$, and aerosol within the boundary layer) also showed an impact on regional air quality. The pollutants emitted above the ocean by ships and oil rigs are thus transported along the West African Monsoon (south-westerly in summer) to the continents (Kniffka et al., 2019; Brocchi et al., 2019), affecting the air quality in those regions. Notably, a pollution plume caused by biomass burning emissions, the aircraft descended to 4.5 km asl."

**MAGIC-2019 project:**

The three flights of this project were designed for in-situ calibration of the SPIRIT instrument, which is already written in the initial manuscript.

**MAGIC-2021 project (two selected flights):**

"On the morning of August 23, 2021, a flight was designed to provide measurements for the validation of satellite products aforementioned in Section 3.7. Intensive vertical and horizontal measurements were carried out, synchronized with several satellite overpasses…

… In addition to long-range transport, local emissions from fires were also observed. Measurements on August 26 showed the impact of two pollution sources. At 9:30 (half an hour after starting the 3.6-hour flight-20210826_1) on August 26, …"

4.  The manuscript needs an improved conclusion and summary section. It would be valuable to highlight which campaign is typical for particular conditions discussed in the manuscript (anthropogenic emissions, clean background air, ship emissions, convection, wildfires, etc.). Please also explain here which campaigns have already been used for either model evaluations or comparisons to satellite measurements.

In addition, the "Summary and Conclusions" section is improved as follows:

**6 Summary and Conclusion**

Thanks to the development of the SPIRIT instrument, accurate airborne CO measurements (0.3 ppbv precision at 1σ and overall uncertainty < 4 ppbv) were achieved during the past decade (2011 – 2021). This database describes all aircraft CO measurements by SPIRIT during this period. More than 200 h of airborne measurements were conducted in Europe, South Asia, and West Africa. The measurement domain covers a wide area, including tropical, northern mid-latitude, and northern high-latitude regions. This database also includes two unique intercontinental measurements, one from Germany to Ghana via the West African coast, and the other one from Germany to Malaysia.

Generally, surface CO levels in Southern Asia, Europe, and the whole Mediterranean region Basin are at a similar level, namely with mixing ratios < 150 ppbv, which is lower than that the coastal region of West Africa (200-300 ppbv). Moreover, CO levels decrease with altitude for most measurements except at high latitude regions where CO increases with altitude. This database can be of particular interest for studying anthropogenic emissions (DACCIWA-2016), aviation emissions (TC2-2013), wildfire emissions (SHIVA-2011, DACCIWA-2016), shipping and offshore oil rig emissions (GLAM-2014, DACCIWA-2016), convective plumes (SHIVA-2011), urban background (ChemCalInt-2014, MAGIC-2019), and long-distance transport (GLAM-2014, MAGIC-2021). This database helps to understand such events and to

constrain and improve relevant model approaches. For instance, based on SHIVA data, it is found that the convective system can significantly affect the composition of the upper troposphere (Krysztofiak et al., 2018; Hamer et al., 2021). Siberia forest fire could affect the atmosphere over the Mediterranean Basin through long-distance transport, which has been confirmed by measurements during the GLAM-2014 project with a combination of a trajectory model and a chemistry-transport model (Brocchi et al., 2018). Brocchi et al. (2019) analyzed DACCIWA measurements and found that offshore oil rigs could contribute to deteriorating the air quality in coastal regions of West Africa. Moreover, with the emerging utilization of satellite images, the importance of calibration of those measurements at a regional and/or a global scale becomes more and more significant, revealing the importance of the SPIRIT database. For example, CO measurements during the DACCIWA-2016 project will be used to validate satellite product from Measurement of Pollution in the Troposphere (MOPITT).

5. It would be helpful for interested readers to also point out sources of other airborne, high quality CO measurements (e.g., US campaigns, balloon-borne measurements etc.).

Yes, it is necessary to point out other airborne CO measurements. We did this in the Introduction (around line 68): "There are already available aircraft CO measurements, which provide a broad spatial distribution of the measurements. Those are mainly from US projects, such as TRACE-P in 2001 (Palmer et al., 2003), INTEX-B in 2006 (Luo et al., 2007), GoAmazon2014/5 in Brazil (Machado et al., 2018), WE-CAN in 2018 (Permar et al., 2021), and FIREX-AQ in 2019 (Bourgeois et al., 2022), and European projects, such as DABEX in 2006 (Johnson et al., 2008), EUCAARI-LONGREX and APPRAISE-ADIENT in 2008 (McMeeking et al., 2010), CLARIFY in 2017 (Haywood et al., 2021), EMeRGe-EU in 2017 and EMeRGe-Asia in 2018 (Forster et al., 2023), BLUESKY in 2020 (Voigt et al., 2022)."

**Minor comments**

6. Line 37-39: Please rewrite sentence, as it is hard to understand.

The sentence has been improved as:

However, those measurements are typically limited to the boundary layer, arising challenges and limitations in understanding atmospheric chemistry and dynamics above this layer (i.e., free troposphere, stratosphere) and showing the necessity of airborne measurements

7. Line 54: Not sure what 'atmospheric dynamics … could be achieved' is supposed to mean.

It is corrected as "understanding the atmospheric dynamics…".

8. Line 63: What target regions are meant here? Is this supposed to be the upper troposphere in general or some very specific regions?

"Target region" means "the studied region". It has been corrected accordingly.

**Comments from Reviewer #2 and Response**

The manuscript written by Xue et al presents a spatial and altitude distribution of carbon monoxide (CO) dataset across the different continents. Overall, the rich airborne CO dataset presented in this manuscript is with high quality and may contribute to the future remote sensing or simulation products development. However, improvements are required in the presentation of the graphs. Additionally, it is advised to have the manuscript thoroughly proofreading before acceptance.

Our response: thank you for your valuable feedback, which contributes to improve the overall quality of our manuscript. The manuscript has been carefully proofreaded again and corrected for English. Please see below the point-to-point response, with your comments in black, our Responses in blue and Changes in red.

All modified Figures are in the Supplement pdf if not clearly visible in the online Reply.

**Major comments:**

1. The presentation of the results and the order of the figures in the manuscript require some improvements: Figure 22 presents an overview of all the flight campaign trajectories, but it comes too late. It would be better to put it at the beginning of the results and discussions section.

We agree with this suggestion. Figure 22 has been moved to the beginning (current Figure 1). And please note that this figure has been significantly modified and improved (see reply to Comment 1 from Reviewer #1).

2. While there is a large amount of the datasets in this study, I recommend the authors to add the flight campaign names in the legend (i.e., SHIVA_20111116) instead of the dates only. Please also describe the measured regions in the figure capitals for Figure 23-29 to help the readers easy to reference.

The legends of all 3D trajectory plots in the main text and the supporting information have been updated by adding the corresponding project names.

Legends of Figures 23-29 have also been updated by adding the corresponding measurement region.

3. Page 10, Line 186: Is shipping emission also a certain type of anthropogenic emission? Please specify the terms more clearly.

Yes, shipping emissions belong to anthropogenic emissions, so "anthropogenic and ship emissions" has been changed to "urban and shipping emissions", now page 12, line 212. Note the "urban emissions" refer to emissions from coastal cities of Cyprus.

4. The authors declare that 'Many polluted plumes are observed'. Please include the measured plume types in the Table 2 to summarize the observed events of different aircraft campaigns.

A similar suggestion is also given by Reviewer #1. Instead of modifying Table 2, we added the potential research interest and corresponding projects in the "Summary and Conclusions" section. Modifications are are as follows:

**6 Summary and Conclusion**

Thanks to the development of the SPIRIT instrument, accurate airborne CO measurements (0.3 ppbv precision at 1σ and overall uncertainty < 4 ppbv) were achieved during the past decade (2011 – 2021). This database describes all aircraft CO measurements by SPIRIT during this period. More than 200 h of airborne measurements were conducted in Europe, South Asia, and West Africa. The measurement domain covers a wide area, including tropical, northern mid-latitude, and northern high-latitude regions. This database also includes two unique intercontinental measurements, one from Germany to Ghana via the West African coast, and the other one from Germany to Malaysia.

Generally, surface CO levels in Southern Asia, Europe, and the whole Mediterranean region Basin are at a similar level, namely with mixing ratios < 150 ppbv, which is lower than that the coastal region of West Africa (200-300 ppbv). Moreover, CO levels decrease with altitude for most measurements except at high latitude regions where CO increases with altitude. This database can be of particular interest for studying anthropogenic emissions (DACCIWA-2016), aviation emissions (TC2-2013), wildfire emissions (SHIVA-2011, DACCIWA-2016), shipping and offshore oil rig emissions (GLAM-2014, DACCIWA-2016), convective plumes (SHIVA-2011), urban background (ChemCalInt-2014, MAGIC-2019), and long-distance transport (GLAM-2014, MAGIC-2021). This database helps to understand such events and to constrain and improve relevant model approaches. For instance, based on SHIVA data, it is found that the convective system can significantly affect the composition of the upper troposphere (Krysztofiak et al., 2018; Hamer et al., 2021). Siberia forest fire could affect the atmosphere over the Mediterranean Basin through long-distance transport, which has been confirmed by measurements during the GLAM-2014 project with a combination of a trajectory model and a chemistry-transport model (Brocchi et al., 2018). Brocchi et al. (2019) analyzed DACCIWA measurements and found that offshore oil rigs could contribute to deteriorating the air quality in coastal regions of West Africa. Moreover, with the emerging utilization of satellite images, the importance of calibration of those measurements at a regional and/or a global scale becomes more and more significant, revealing the importance of the SPIRIT database. For example, CO measurements during the DACCIWA-2016 project will be used to validate satellite product from Measurement of Pollution in the Troposphere (MOPITT).

5. For the conclusion section, please also briefly summarize some key results of the altitude profile measurements to highlight the importance of this dataset.

Yes, it is necessary to do that. We added two sentences in the Section of Summary and Conclusion:

Generally, surface CO levels in Southern Asia, Europe, and the whole Mediterranean region Basin are at a similar level, namely with mixing ratios < 150 ppbv, which are lower than that the coastal region of West Africa (200-300 ppbv). Moreover, CO levels decrease with altitude for most measurements except at high latitude regions where CO increases with altitude. This database can be of particular interest for studying anthropogenic emissions (DACCIWA-2016), aviation emissions (TC2-2013), wildfire emissions (SHIVA-2011, DACCIWA-2016), shipping and offshore oil rig emissions (GLAM-2014, DACCIWA-2016), convective plumes (SHIVA-2011), urban background (ChemCalInt-2014, MAGIC-2019), and long-distance transport (GLAM-2014, MAGIC-2021).

**Minor comments:**

**6.** The Google Map resolution of Figure 3 and Figure 22 is poor, I encourage the authors to regenerate the figures.

Improved as suggested. Please note that after re-ordering the figures, Figures 3 and 22 are Figures 4 and 1, respectively.

[Figure]

**Figure 1. Locations of the main airports used for each project. Map copyright: © GoogleMap. The same airport in Toulouse was used for TC2-2013, ChemCalInt-2014, and Test ATR-2016, then the overlapped points are offset and plotted horizontally. Lampedusa E Linosa airport is selected for GLAM-2014 as it is in the center of the measurement area. Barplots show the average CO levels in the boundary layer (0-2 km), lower free troposphere (2-6 km) and upper free troposphere (>6 km) for each project.**

[Figure]

**Figure 4: Flight trajectories during the TC2 2013 campaign. Map copyright: © GoogleMap.**

7. Page 1, Line 18: Do you mean 'spatial and vertical distribution'?

No, we meant tridimensional, so corrected as "spatial".

8. Page 2, Line 42: 'many important gases are important for atmospheric chemistry': please rephrase the duplicated statement here.

The second "important" has been deleted and the sentence has been improved as "many important gases for atmospheric chemistry, air quality, and global climate have too low abundance to be detected by satellite."

9. Page 2, Line 54: Not sure what is the meaning of the term 'atmospheric dynamics and the distribution of pollution' here.

The sentence has been corrected as "understanding the atmospheric dynamics and the distribution…".

10. Page 11, Line 197: 'the sample inlet is connected to the standard', what is the meaning of 'standard'?

It has been improved as "Each time the sampling inlet is connected to the cylinder containing the WMO CO standard"

11. Page 13, Line 207, there are couples of aircraft campaigns conducted in West Africa: Formenti et al (2019), Haywood et al (2021), Redemann et al (2021), and Wu et al (2021)

Thanks for the suggestion. These references are therefore cited in the manuscript.

12. Page 15, Line 245, 'ascending, descending, and cruise periods' ?

Yes, "period", so corrected accordingly.